# Gradient Estimators for Implicit Models

**Yingzhen Li & Richard E. Turner**
University of Cambridge
Cambridge, CB2 1PZ, UK
`{yl494,ret26}@cam.ac.uk`

## Abstract

Implicit models, which allow for the generation of samples but not for point-wise evaluation of probabilities, are omnipresent in real-world problems tackled by machine learning and a hot topic of current research. Some examples include data simulators that are widely used in engineering and scientific research, generative adversarial networks (GANs) for image synthesis, and hot-off-the-press approximate inference techniques relying on implicit distributions. The majority of existing approaches to learning implicit models rely on approximating the intractable distribution or optimisation objective for gradient-based optimisation, which is liable to produce inaccurate updates and thus poor models. This paper alleviates the need for such approximations by proposing the *Stein gradient estimator*, which directly estimates the score function of the implicitly defined distribution. The efficacy of the proposed estimator is empirically demonstrated by examples that include gradient-free MCMC, meta-learning for approximate inference and entropy regularised GANs that provide improved sample diversity.

## 1 Introduction

Modelling is fundamental to the success of technological innovations for artificial intelligence. A powerful model learns a useful representation of the observations for a specified prediction task, and generalises to unknown instances that follow similar generative mechanics. A well established area of machine learning research focuses on developing *prescribed probabilistic models* (Diggle & Gratton, 1984), where learning is based on evaluating the probability of observations under the model. *Implicit probabilistic models*, on the other hand, are defined by a stochastic procedure that allows for direct generation of samples, but not for the evaluation of model probabilities. These are omnipresent in scientific and engineering research involving data analysis, for instance ecology, climate science and geography, where simulators are used to fit real-world observations to produce forecasting results. Within the machine learning community there is a recent interest in a specific type of implicit models, generative adversarial networks (GANs) (Goodfellow et al., 2014), which has been shown to be one of the most successful approaches to image and text generation (Radford et al., 2016; Yu et al., 2017; Arjovsky et al., 2017; Berthelot et al., 2017). Very recently, implicit distributions have also been considered as approximate posterior distributions for Bayesian inference, e.g. see Liu & Feng (2016); Wang & Liu (2016); Li & Liu (2016); Karaletsos (2016); Mescheder et al. (2017); Huszár (2017); Li et al. (2017); Tran et al. (2017). These examples demonstrate the superior flexibility of implicit models, which provide highly expressive means of modelling complex data structures.

Whilst prescribed probabilistic models can be learned by standard (approximate) maximum likelihood or Bayesian inference, implicit probabilistic models require substantially more severe approximations due to the intractability of the model distribution. Many existing approaches first approximate the model distribution or optimisation objective function and then use those approximations to learn the associated parameters. However, for any finite number of data points there exists an infinite number of functions, with arbitrarily diverse gradients, that can approximate perfectly the objective function at the training datapoints, and optimising such approximations can lead to unstable training and poor results. Recent research on GANs, where the issue is highly prevalent, suggest that restricting the representational power of the discriminator is effective in stabilising training (e.g. see Arjovsky et al., 2017; Kodali et al., 2017). However, such restrictions often intro-

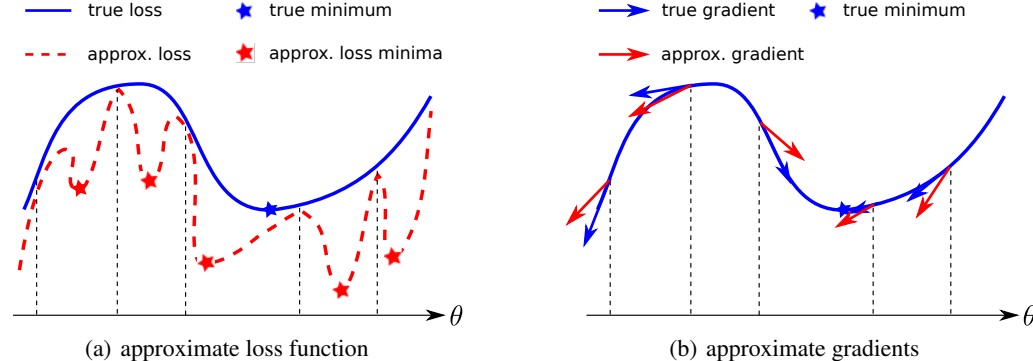

Figure 1: A comparison between the two approximation schemes. Since in practice the optimiser only visits finite number of locations in the parameter space, it can lead to over-fitting if the neural network based functional approximator is not carefully regularised, and therefore the curvature information of the approximated loss can be very different from that of the original loss (shown in (a)). On the other hand, the gradient approximation scheme (b) can be more accurate since it only involves estimating the sensitivity of the loss function to the parameters in a local region.

duce undesirable biases, responsible for problems such as mode collapse in the context of GANs, and the underestimation of uncertainty in variational inference methods (Turner & Sahani, 2011).

In this paper we explore approximating the derivative of the log density, known as the score function, as an alternative method for training implicit models. An accurate approximation of the score function then allows the application of many well-studied algorithms, such as maximum likelihood, maximum entropy estimation, variational inference and gradient-based MCMC, to implicit models. Concretely, our contributions include:

- the *Stein gradient estimator*, a novel generalisation of the score matching gradient estimator (Hyvärinen, 2005), that includes both parametric and non-parametric forms;
- a comparison of the proposed estimator with the score matching and the KDE plug-in estimators on performing gradient-free MCMC, meta-learning of approximate posterior samplers for Bayesian neural networks, and entropy based regularisation of GANs.

## 2 LEARNING IMPLICIT PROBABILISTIC MODELS

Given a dataset $\mathcal{D}$ containing i.i.d. samples we would like to learn a probabilistic model $p(\boldsymbol{x})$ for the underlying data distribution $p_{\mathcal{D}}(\boldsymbol{x})$. In the case of implicit models, $p(\boldsymbol{x})$ is defined by a generative process. For example, to generate images, one might define a generative model $p(\boldsymbol{x})$ that consists of sampling randomly a latent variable $\boldsymbol{z} \sim p_0(\boldsymbol{z})$ and then defining $\boldsymbol{x} = \boldsymbol{f}_{\boldsymbol{\theta}}(\boldsymbol{z})$. Here $\boldsymbol{f}$ is a function parametrised by $\boldsymbol{\theta}$, usually a deep neural network or a simulator. We assume $\boldsymbol{f}$ to be differentiable w.r.t. $\boldsymbol{\theta}$. An extension to this scenario is presented by *conditional* implicit models, where the addition of a supervision signal $\boldsymbol{y}$, such as an image label, allows us to define a conditional distribution $p(\boldsymbol{x}|\boldsymbol{y})$ implicitly by the transformation $\boldsymbol{x} = \boldsymbol{f}_{\boldsymbol{\theta}}(\boldsymbol{z}, \boldsymbol{y})$. A related methodology, *wild variational inference* (Liu & Feng, 2016; Li & Liu, 2016) assumes a tractable joint density $p(\boldsymbol{x}, \boldsymbol{z})$, but uses implicit proposal distributions to approximate an intractable exact posterior $p(\boldsymbol{z}|\boldsymbol{x})$. Here the approximate posterior $q(\boldsymbol{z}|\boldsymbol{x})$ can likewise be represented by a deep neural network, but also by a truncated Markov chain, such as that given by Langevin dynamics with learnable step-size.

Whilst providing extreme flexibility and expressive power, the intractability of density evaluation also brings serious optimisation issues for implicit models. This is because many learning algorithms, e.g. maximum likelihood estimation (MLE), rely on minimising a distance/divergence/discrepancy measure $\mathrm{D}[p||p_{\mathcal{D}}]$, which often requires evaluating the model density (c.f. Ranganath et al., 2016; Liu & Feng, 2016). Thus good approximations to the optimisation procedure are the key to learning implicit models that can describe complex data structure. In the context of GANs, the Jensen-Shannon divergence is approximated by a variational lower-bound

represented by a discriminator (Barber & Agakov, 2003; Goodfellow et al., 2014). Related work for wild variational inference (Li & Liu, 2016; Mescheder et al., 2017; Huszár, 2017; Tran et al., 2017) uses a GAN-based technique to construct a density ratio estimator for $q/p_0$ (Sugiyama et al., 2009; 2012; Uehara et al., 2016; Mohamed & Lakshminarayanan, 2016) and then approximates the KL-divergence term in the variational lower-bound:

$$\mathcal{L}_{\text{VI}}(q) = \mathbb{E}_q \left[ \log p(\boldsymbol{x}|\boldsymbol{z}) \right] - \text{KL}[q_{\boldsymbol{\phi}}(\boldsymbol{z}|\boldsymbol{x}) || p_0(\boldsymbol{z})]. \tag{1}$$

In addition, Li & Liu (2016) and Mescheder et al. (2017) exploit the additive structure of the KL-divergence and suggest discriminating between $q$ and an auxiliary distribution that is close to $q$, making the density ratio estimation more accurate. Nevertheless all these algorithms involve a min-imax optimisation, and the current practice of gradient-based optimisation is notoriously unstable.

The stabilisation of GAN training is itself a recent trend of related research (e.g. see Salimans et al., 2016; Arjovsky et al., 2017). However, as the gradient-based optimisation only interacts with gradients, there is no need to use a discriminator if an accurate approximation to the intractable gradients could be obtained. As an example, consider a variational inference task with the approximate posterior defined as $\boldsymbol{z} \sim q_{\boldsymbol{\phi}}(\boldsymbol{z}|\boldsymbol{x}) \Leftrightarrow \boldsymbol{\epsilon} \sim \pi(\boldsymbol{\epsilon}), \boldsymbol{z} = \boldsymbol{f}_{\boldsymbol{\phi}}(\boldsymbol{\epsilon}, \boldsymbol{x})$. Notice that the variational lower-bound can be rewritten as

$$\mathcal{L}_{\text{VI}}(q) = \mathbb{E}_q \left[ \log p(\boldsymbol{x}, \boldsymbol{z}) \right] + \mathbb{H}[q_{\boldsymbol{\phi}}(\boldsymbol{z}|\boldsymbol{x})], \tag{2}$$

the gradient of the variational parameters $\boldsymbol{\phi}$ can be computed by a sum of the path gradient of the first term (i.e. $\mathbb{E}_\pi \left[ \nabla_{\boldsymbol{f}} \log p(\boldsymbol{x}, \boldsymbol{f}(\boldsymbol{\epsilon}, \boldsymbol{x}))^{\text{T}} \nabla_{\boldsymbol{\phi}} \boldsymbol{f}(\boldsymbol{\epsilon}, \boldsymbol{x}) \right]$) and the gradient of the entropy term $\nabla_{\boldsymbol{\phi}} \mathbb{H}[q(\boldsymbol{z}|\boldsymbol{x})]$. Expanding the latter, we have

$$\begin{aligned}
\nabla_{\boldsymbol{\phi}} \mathbb{H}[q_{\boldsymbol{\phi}}(\boldsymbol{z}|\boldsymbol{x})] &= -\nabla_{\boldsymbol{\phi}} \mathbb{E}_{\pi(\boldsymbol{\epsilon})}[\log q_{\boldsymbol{\phi}}(\boldsymbol{f}_{\boldsymbol{\phi}}(\boldsymbol{\epsilon}, \boldsymbol{x}))] \\
&= -\mathbb{E}_{\pi(\boldsymbol{\epsilon})}[\nabla_{\boldsymbol{\phi}} \log q_{\boldsymbol{\phi}}(\boldsymbol{f}_{\boldsymbol{\phi}}(\boldsymbol{\epsilon}, \boldsymbol{x}))] \\
&= -\mathbb{E}_{\pi(\boldsymbol{\epsilon})}[\nabla_{\boldsymbol{\phi}} \log q_{\boldsymbol{\phi}}(\boldsymbol{z}|\boldsymbol{x})|_{\boldsymbol{z}=\boldsymbol{f}_{\boldsymbol{\phi}}(\boldsymbol{\epsilon}, \boldsymbol{x})} + \nabla_{\boldsymbol{f}} \log q_{\boldsymbol{\phi}}(\boldsymbol{f}_{\boldsymbol{\phi}}(\boldsymbol{\epsilon}, \boldsymbol{x})|\boldsymbol{x}) \nabla_{\boldsymbol{\phi}} \boldsymbol{f}_{\boldsymbol{\phi}}(\boldsymbol{\epsilon}, \boldsymbol{x})] \\
&= -\mathbb{E}_{q_{\boldsymbol{\phi}}(\boldsymbol{z}|\boldsymbol{x})}[\nabla_{\boldsymbol{\phi}} \log q_{\boldsymbol{\phi}}(\boldsymbol{z}|\boldsymbol{x})] - \mathbb{E}_{\pi(\boldsymbol{\epsilon})}[\nabla_{\boldsymbol{f}} \log q_{\boldsymbol{\phi}}(\boldsymbol{f}_{\boldsymbol{\phi}}(\boldsymbol{\epsilon}, \boldsymbol{x})|\boldsymbol{x}) \nabla_{\boldsymbol{\phi}} \boldsymbol{f}_{\boldsymbol{\phi}}(\boldsymbol{\epsilon}, \boldsymbol{x})],
\end{aligned} \tag{3}$$

in which the first term in the last line is zero (Roeder et al., 2017). As we typically assume the tractability of $\nabla_{\boldsymbol{\phi}} \boldsymbol{f}$, an accurate approximation to $\nabla_{\boldsymbol{z}} \log q(\boldsymbol{z}|\boldsymbol{x})$ would remove the requirement of discriminators, speed-up the learning and obtain potentially a better model. Many gradient approximation techniques exist (Stone, 1985; Fan & Gijbels, 1996; Zhou & Wolfe, 2000; De Brabanter et al., 2013), and in particular, in the next section we will review kernel-based methods such as kernel density estimation (Singh, 1977) and score matching (Hyvärinen, 2005) in more detail, and motivate the main contribution of the paper.

## 3   GRADIENT APPROXIMATION WITH THE STEIN GRADIENT ESTIMATOR

We propose the *Stein gradient estimator* as a novel generalisation of the score matching gradient estimator. Before presenting it we first set-up the notation. Column vectors and matrices are boldfaced. The random variable under consideration is $\boldsymbol{x} \in \mathcal{X}$ with $\mathcal{X} = \mathbb{R}^{d \times 1}$ if not specifically mentioned. To avoid misleading notation we use the distribution $q(\boldsymbol{x})$ to derive the gradient approximations for general cases. As Monte Carlo methods are heavily used for implicit models, in the rest of the paper we mainly consider approximating the gradient $\boldsymbol{g}(\boldsymbol{x}^k) := \nabla_{\boldsymbol{x}^k} \log q(\boldsymbol{x}^k)$ for $\boldsymbol{x}^k \sim q(\boldsymbol{x}), k = 1, ..., K$. We use $x_j^i$ to denote the $j$th element of the $i$th sample $\boldsymbol{x}^i$. We also denote the matrix form of the collected gradients as $\mathbf{G} := \left( \nabla_{\boldsymbol{x}^1} \log q(\boldsymbol{x}^1), \cdots, \nabla_{\boldsymbol{x}^K} \log q(\boldsymbol{x}^K) \right)^{\text{T}} \in \mathbb{R}^{K \times d}$, and its approximation $\hat{\mathbf{G}} := \left( \hat{g}(\boldsymbol{x}^1), \cdots, \hat{g}(\boldsymbol{x}^K) \right)^{\text{T}}$ with $\hat{g}(\boldsymbol{x}^k) = \nabla_{\boldsymbol{x}^k} \log \hat{q}(\boldsymbol{x}^k)$ for some $\hat{q}(\boldsymbol{x})$.

### 3.1   STEIN GRADIENT ESTIMATOR: INVERTING STEIN'S IDENTITY

We start from introducing *Stein's identity* that was first developed for Gaussian random variables (Stein, 1972; 1981) then extended to general cases (Gorham & Mackey, 2015; Liu et al., 2016). Let $\boldsymbol{h} : \mathbb{R}^{d \times 1} \to \mathbb{R}^{d' \times 1}$ be a differentiable multivariate test function which maps $\boldsymbol{x}$ to a column vector $\boldsymbol{h}(\boldsymbol{x}) = [h_1(\boldsymbol{x}), h_2(\boldsymbol{x}), ..., h_{d'}(\boldsymbol{x})]^{\text{T}}$. We further assume the *boundary condition* for $\boldsymbol{h}$:

$$q(\boldsymbol{x})\boldsymbol{h}(\boldsymbol{x})|_{\partial \mathcal{X}} = \mathbf{0}, \text{ or } \lim_{\boldsymbol{x} \to \infty} q(\boldsymbol{x})\boldsymbol{h}(\boldsymbol{x}) = 0 \text{ if } \mathcal{X} = \mathbb{R}^d. \tag{4}$$

This condition holds for almost any test function if $q$ has sufficiently fast-decaying tails (e.g. Gaussian tails). Now we introduce Stein's identity (Stein, 1981; Gorham & Mackey, 2015; Liu et al., 2016)

$$\mathbb{E}_q[\boldsymbol{h}(\boldsymbol{x})\nabla_{\boldsymbol{x}}\log q(\boldsymbol{x})^{\mathrm{T}} + \nabla_{\boldsymbol{x}}\boldsymbol{h}(\boldsymbol{x})] = \boldsymbol{0}, \tag{5}$$

in which the gradient matrix term $\nabla_{\boldsymbol{x}}\boldsymbol{h}(\boldsymbol{x}) = (\nabla_{\boldsymbol{x}}h_1(\boldsymbol{x}), \cdots, \nabla_{\boldsymbol{x}}h_{d'}(\boldsymbol{x}))^{\mathrm{T}} \in \mathbb{R}^{d' \times d}$. This identity can be proved using *integration by parts*: for the $i$th row of the matrix $\boldsymbol{h}(\boldsymbol{x})\nabla_{\boldsymbol{x}}\log q(\boldsymbol{x})^{\mathrm{T}}$, we have

$$\begin{aligned}
\mathbb{E}_q[h_i(\boldsymbol{x})\nabla_{\boldsymbol{x}}\log q(\boldsymbol{x})^{\mathrm{T}}] &= \int h_i(\boldsymbol{x})\nabla_{\boldsymbol{x}}q(\boldsymbol{x})^{\mathrm{T}}d\boldsymbol{x} \\
&= q(\boldsymbol{x})h_i(\boldsymbol{x})|_{\partial\mathcal{X}} - \int q(\boldsymbol{x})\nabla_{\boldsymbol{x}}h_i(\boldsymbol{x})^{\mathrm{T}}d\boldsymbol{x} \\
&= -\mathbb{E}_q[\nabla_{\boldsymbol{x}}h_i(\boldsymbol{x})^{\mathrm{T}}].
\end{aligned} \tag{6}$$

Observing that the gradient term $\nabla_{\boldsymbol{x}}\log q(\boldsymbol{x})$ of interest appears in Stein's identity (5), we propose the *Stein gradient estimator* by inverting Stein's identity. As the expectation in (5) is intractable, we further approximate the above with Monte Carlo (MC):

$$\frac{1}{K}\sum_{k=1}^{K} -\boldsymbol{h}(\boldsymbol{x}^k)\nabla_{\boldsymbol{x}^k}\log q(\boldsymbol{x}^k)^{\mathrm{T}} + \mathrm{err} = \frac{1}{K}\sum_{k=1}^{K}\nabla_{\boldsymbol{x}^k}\boldsymbol{h}(\boldsymbol{x}^k), \quad \boldsymbol{x}^k \sim q(\boldsymbol{x}^k), \tag{7}$$

with $\mathrm{err} \in \mathbb{R}^{d' \times d}$ the random error due to MC approximation, which has mean $\boldsymbol{0}$ and vanishes as $K \to +\infty$. Now by temporarily denoting $\mathbf{H} = (\boldsymbol{h}(\boldsymbol{x}^1), \cdots, \boldsymbol{h}(\boldsymbol{x}^K)) \in \mathbb{R}^{d' \times K}$, $\overline{\nabla_{\boldsymbol{x}}\boldsymbol{h}} = \frac{1}{K}\sum_{k=1}^{K}\nabla_{\boldsymbol{x}^k}\boldsymbol{h}(\boldsymbol{x}^k) \in \mathbb{R}^{d' \times d}$, equation (7) can be rewritten as $-\frac{1}{K}\mathbf{H}\mathbf{G} + \mathrm{err} = \overline{\nabla_{\boldsymbol{x}}\boldsymbol{h}}$. Thus we consider a ridge regression method (i.e. adding an $\ell_2$ regulariser) to estimate $\mathbf{G}$:

$$\hat{\mathbf{G}}_V^{\mathrm{Stein}} := \underset{\hat{\mathbf{G}}\in\mathbb{R}^{K\times d}}{\arg\min} ||\overline{\nabla_{\boldsymbol{x}}\boldsymbol{h}} + \frac{1}{K}\mathbf{H}\hat{\mathbf{G}}||_F^2 + \frac{\eta}{K^2}||\hat{\mathbf{G}}||_F^2, \tag{8}$$

with $||\cdot||_F$ the Frobenius norm of a matrix and $\eta \geq 0$. Simple calculation shows that

$$\hat{\mathbf{G}}_V^{\mathrm{Stein}} = -(\mathbf{K} + \eta\boldsymbol{I})^{-1}\langle\nabla, \mathbf{K}\rangle, \tag{9}$$

where $\mathbf{K} := \mathbf{H}^{\mathrm{T}}\mathbf{H}$, $\mathbf{K}_{ij} = \mathcal{K}(\boldsymbol{x}^i, \boldsymbol{x}^j) := \boldsymbol{h}(\boldsymbol{x}^i)^{\mathrm{T}}\boldsymbol{h}(\boldsymbol{x}^j)$, $\langle\nabla, \mathbf{K}\rangle := K\mathbf{H}^{\mathrm{T}}\overline{\nabla_{\boldsymbol{x}}\boldsymbol{h}}$, $\langle\nabla, \mathbf{K}\rangle_{ij} = \sum_{k=1}^{K}\nabla_{x_j^k}\mathcal{K}(\boldsymbol{x}^i, \boldsymbol{x}^k)$. One can show that the RBF kernel satisfies Stein's identity (Liu et al., 2016). In this case $\boldsymbol{h}(\boldsymbol{x}) = \mathcal{K}(\boldsymbol{x}, \cdot)$, $d' = +\infty$ and by the reproducing kernel property (Berlinet & Thomas-Agnan, 2011), $\boldsymbol{h}(\boldsymbol{x})^{\mathrm{T}}\boldsymbol{h}(\boldsymbol{x}') = \langle\mathcal{K}(\boldsymbol{x}, \cdot), \mathcal{K}(\boldsymbol{x}', \cdot)\rangle_{\mathcal{H}} = \mathcal{K}(\boldsymbol{x}, \boldsymbol{x}')$.

### 3.2 STEIN GRADIENT ESTIMATOR MINIMISES THE KERNELISED STEIN DISCREPANCY

In this section we derive the Stein gradient estimator again, but from a divergence/discrepancy minimisation perspective. Stein's method also provides a tool for checking if two distributions $q(\boldsymbol{x})$ and $\hat{q}(\boldsymbol{x})$ are identical. If the test function set $\mathcal{H}$ is sufficiently rich, then one can define a Stein discrepancy measure by

$$\mathcal{S}(q, \hat{q}) := \sup_{\boldsymbol{h}\in\mathcal{H}}\mathbb{E}_q\left[\nabla_{\boldsymbol{x}}\log\hat{q}(\boldsymbol{x})^{\mathrm{T}}\boldsymbol{h}(\boldsymbol{x}) + \langle\nabla, \boldsymbol{h}\rangle\right], \tag{10}$$

see Gorham & Mackey (2015) for an example derivation. When $\mathcal{H}$ is defined as a unit ball in an RKHS induced by a kernel $\mathcal{K}(\boldsymbol{x}, \cdot)$, Liu et al. (2016) and Chwialkowski et al. (2016) showed that the supremum in (10) can be analytically obtained as (with $\mathcal{K}_{\boldsymbol{xx}'}$ shorthand for $\mathcal{K}(\boldsymbol{x}, \boldsymbol{x}')$):

$$\mathcal{S}^2(q, \hat{q}) = \mathbb{E}_{\boldsymbol{x},\boldsymbol{x}'\sim q}\left[(\hat{\boldsymbol{g}}(\boldsymbol{x}) - \boldsymbol{g}(\boldsymbol{x}))^{\mathrm{T}}\mathcal{K}_{\boldsymbol{xx}'}(\hat{\boldsymbol{g}}(\boldsymbol{x}') - \boldsymbol{g}(\boldsymbol{x}'))\right], \tag{11}$$

which is also named the *kernelised Stein discrepancy* (KSD). Chwialkowski et al. (2016) showed that for $C_0$-universal kernels satisfying the boundary condition, KSD is indeed a discrepancy measure: $\mathcal{S}^2(q, \hat{q}) = 0 \Leftrightarrow q = \hat{q}$. Gorham & Mackey (2017) further characterised the power of KSD on detecting non-convergence cases. Furthermore, if the kernel is twice differentiable, then using the same technique as to derive (16) one can compute KSD by

$$\mathcal{S}^2(q, \hat{q}) = \mathbb{E}_{\boldsymbol{x},\boldsymbol{x}'\sim q}\left[\hat{\boldsymbol{g}}(\boldsymbol{x})^{\mathrm{T}}\mathcal{K}_{\boldsymbol{xx}'}\hat{\boldsymbol{g}}(\boldsymbol{x}') + \hat{\boldsymbol{g}}(\boldsymbol{x})^{\mathrm{T}}\nabla_{\boldsymbol{x}'}\mathcal{K}_{\boldsymbol{xx}'} + \nabla_{\boldsymbol{x}}\mathcal{K}_{\boldsymbol{xx}'}^{\mathrm{T}}\hat{\boldsymbol{g}}(\boldsymbol{x}') + \mathrm{Tr}(\nabla_{\boldsymbol{x},\boldsymbol{x}'}\mathcal{K}_{\boldsymbol{xx}'})\right]. \tag{12}$$

In practice KSD is estimated with samples $\{\boldsymbol{x}^k\}_{k=1}^{K} \sim q$, and simple derivations show that the V-statistic of KSD can be reformulated as $\mathcal{S}_V^2(q, \hat{q}) = \frac{1}{K^2}\mathrm{Tr}(\hat{\mathbf{G}}^{\mathrm{T}}\mathbf{K}\hat{\mathbf{G}} + 2\hat{\mathbf{G}}^{\mathrm{T}}\langle\nabla, \mathbf{K}\rangle) + C$. Thus the $l_2$ error in (8) is equivalent to the V-statistic of KSD if $\boldsymbol{h}(\boldsymbol{x}) = \mathcal{K}(\boldsymbol{x}, \cdot)$, and we have the following:

**Theorem 1.** $\hat{\mathbf{G}}_V^{\text{Stein}}$ *is the solution of the following KSD V-statistic minimisation problem*

$$\hat{\mathbf{G}}_V^{\text{Stein}} = \underset{\hat{\mathbf{G}} \in \mathbb{R}^{K \times d}}{\arg\min} \mathcal{S}_V^2(q, \hat{q}) + \frac{\eta}{K^2} ||\hat{\mathbf{G}}||_F^2. \tag{13}$$

One can also minimise the U-statistic of KSD to obtain gradient approximations, and a full derivation of which, including the optimal solution, can be found in the appendix. In experiments we use V-statistic solutions and leave comparisons between these methods to future work.

### 3.3 COMPARISONS TO EXISTING KERNEL-BASED GRADIENT ESTIMATORS

There exist other gradient estimators that do not require explicit evaluations of $\nabla_{\boldsymbol{x}} \log q(\boldsymbol{x})$, e.g. the denoising auto-encoder (DAE) (Vincent et al., 2008; Vincent, 2011; Alain & Bengio, 2014) which, with infinitesimal noise, also provides an estimate of $\nabla_{\boldsymbol{x}} \log q(\boldsymbol{x})$ at convergence. However, applying such gradient estimators result in a double-loop optimisation procedure since the gradient approximation is repeatedly required for fitting implicit distributions, which can be significantly slower than the proposed approach. Therefore we focus on "quick and dirty" approximations and only include comparisons to kernel-based gradient estimators in the following.

#### 3.3.1 KDE GRADIENT ESTIMATOR: PLUG-IN ESTIMATOR WITH DENSITY ESTIMATION

A naive approach for gradient approximation would first estimate the intractable density $\hat{q}(\boldsymbol{x}) \approx q(\boldsymbol{x})$ (up to a constant), then approximate the exact gradient by $\nabla_{\boldsymbol{x}} \log \hat{q}(\boldsymbol{x}) \approx \nabla_{\boldsymbol{x}} \log q(\boldsymbol{x})$. Specifically, Singh (1977) considered kernel density estimation (KDE) $\hat{q}(\boldsymbol{x}) = \frac{1}{K} \sum_{k=1}^{K} \mathcal{K}(\boldsymbol{x}, \boldsymbol{x}^k) \times C.$, then differentiated through the KDE estimate to obtain the gradient estimator:

$$\hat{\mathbf{G}}_{ij}^{\text{KDE}} = \sum_{k=1}^{K} \nabla_{x_j^i} \mathcal{K}(\boldsymbol{x}^i, \boldsymbol{x}^k) / \sum_{k=1}^{K} \mathcal{K}(\boldsymbol{x}^i, \boldsymbol{x}^k). \tag{14}$$

Interestingly for translation invariant kernels $\mathcal{K}(\boldsymbol{x}, \boldsymbol{x}') = \mathcal{K}(\boldsymbol{x} - \boldsymbol{x}')$ the *KDE gradient estimator* (14) can be rewritten as $\hat{\mathbf{G}}^{\text{KDE}} = -\text{diag}\left(\mathbf{K}\mathbf{1}\right)^{-1} \langle \nabla, \mathbf{K} \rangle$. Inspecting and comparing it with the Stein gradient estimator (9), one might notice that the Stein method uses the full kernel matrix as the pre-conditioner, while the KDE method computes an averaged "kernel similarity" for the denominator. We conjecture that this difference is key to the superior performance of the Stein gradient estimator when compared to the KDE gradient estimator (see later experiments). The KDE method only collects the similarity information between $\boldsymbol{x}^k$ and other samples $\boldsymbol{x}^j$ to form an estimate of $\nabla_{\boldsymbol{x}^k} \log q(\boldsymbol{x}^k)$, whereas for the Stein gradient estimator, the kernel similarity between $\boldsymbol{x}^i$ and $\boldsymbol{x}^j$ for all $i, j \neq k$ are also incorporated. Thus it is reasonable to conjecture that the Stein method can be more sample efficient, which also implies higher accuracy when the same number of samples are collected.

#### 3.3.2 SCORE MATCHING GRADIENT ESTIMATOR: MINIMISING MSE

The KDE gradient estimator performs indirect approximation of the gradient via density estimation, which can be inaccurate. An alternative approach directly approximates the gradient $\nabla_{\boldsymbol{x}} \log q(\boldsymbol{x})$ by minimising the expected $\ell_2$ error w.r.t. the approximation $\hat{\boldsymbol{g}}(\boldsymbol{x}) = (\hat{g}_1(\boldsymbol{x}), \cdots, \hat{g}_d(\boldsymbol{x}))^{\text{T}}$:

$$\mathcal{F}(\hat{\boldsymbol{g}}) := \mathbb{E}_q \left[ ||\hat{\boldsymbol{g}}(\boldsymbol{x}) - \nabla_{\boldsymbol{x}} \log q(\boldsymbol{x})||_2^2 \right]. \tag{15}$$

It has been shown in Hyvärinen (2005) that this objective can be reformulated as

$$\mathcal{F}(\hat{\boldsymbol{g}}) = \mathbb{E}_q \left[ ||\hat{\boldsymbol{g}}(\boldsymbol{x})||_2^2 + 2\langle \nabla, \hat{\boldsymbol{g}}(\boldsymbol{x}) \rangle \right] + C, \quad \langle \nabla, \hat{\boldsymbol{g}}(\boldsymbol{x}) \rangle = \sum_{j=1}^{d} \nabla_{x_j} \hat{g}_j(\boldsymbol{x}). \tag{16}$$

The key insight here is again the usage of integration by parts: after expanding the $\ell_2$ loss objective, the cross term can be rewritten as $\mathbb{E}_q \left[ \hat{\boldsymbol{g}}(\boldsymbol{x})^{\text{T}} \nabla_{\boldsymbol{x}} \log q(\boldsymbol{x}) \right] = -\mathbb{E}_q \left[ \langle \nabla, \hat{\boldsymbol{g}}(\boldsymbol{x}) \rangle \right]$, if assuming the boundary condition (4) for $\hat{\boldsymbol{g}}$ (see (6)). The optimum of (16) is referred as the *score matching gradient estimator*. The $\ell_2$ objective (15) is also called *Fisher divergence* (Johnson, 2004) which is a special case of KSD (11) by selecting $\mathcal{K}(\boldsymbol{x}, \boldsymbol{x}') = \delta_{\boldsymbol{x} = \boldsymbol{x}'}$. Thus the Stein gradient estimator can be viewed as a generalisation of the score matching estimator.

The comparison between the two estimators is more complicated. Certainly by the Cauchy-Schwarz inequality the Fisher divergence is stronger than KSD in terms of detecting convergence (Liu et al., 2016). However it is difficult to perform direct gradient estimation by minimising the Fisher divergence, since (i) the Dirac kernel is non-differentiable so that it is impossible to rewrite the divergence in a similar form to (12), and (ii) the transformation to (16) involves computing $\nabla_{\boldsymbol{x}} \hat{\boldsymbol{g}}(\boldsymbol{x})$. So one needs to propose a *parametric* approximation to $\mathbf{G}$ and then optimise the associated parameters accordingly, and indeed Sasaki et al. (2014) and Strathmann et al. (2015) derived a parametric solution by first approximating the log density up to a constant as $\log \hat{q}(\boldsymbol{x}) := \sum_{k=1}^{K} a_k \mathcal{K}(\boldsymbol{x}, \boldsymbol{x}^k) + C$, then minimising (16) to obtain the coefficients $\hat{a}_k^{\text{score}}$ and constructing the gradient estimator as

$$\hat{\mathbf{G}}_{i\cdot}^{\text{score}} = \sum_{k=1}^{K} \hat{a}_k^{\text{score}} \nabla_{\boldsymbol{x}^i} \mathcal{K}(\boldsymbol{x}^i, \boldsymbol{x}^k). \tag{17}$$

Therefore the usage of parametric estimation can potentially remove the advantage of using a stronger divergence. Conversely, the proposed Stein gradient estimator (9) is *non-parametric* in that it directly optimises over functions evaluated at locations $\{\boldsymbol{x}_k\}_{k=1}^{K}$. This brings in two key advantages over the score matching gradient estimator: (i) it removes the *approximation error* due to the use of restricted family of parametric approximations and thus can be potentially more accurate; (ii) it has a much simpler and *ubiquitous* form that applies to *any kernel satisfying the boundary condition*, whereas the score matching estimator requires tedious derivations for different kernels repeatedly (see appendix).

In terms of computation speed, since in most of the cases the computation of the score matching gradient estimator also involves kernel matrix inversions, both estimators are of the same order of complexity, which is $\mathcal{O}(K^3 + K^2 d)$ (kernel matrix computation plus inversion). Low-rank approximations such as the Nyström method (Smola & Schökopf, 2000; Williams & Seeger, 2001) can enable speed-up, but this is not investigated in the paper. Again we note here that kernel-based gradient estimators can still be faster than e.g. the DAE estimator since no double-loop optimisation is required. Certainly it is possible to apply early-stopping for the inner-loop DAE fitting. However the resulting gradient approximation might be very poor, which leads to unstable training and poorly fitted implicit distributions.

### 3.4 Adding predictive power

Though providing potentially more accurate approximations, the non-parametric estimator (9) has no predictive power as described so far. Crucially, many tasks in machine learning require predicting gradient functions at samples drawn from distributions other than $q$, for example, in MLE $q(\boldsymbol{x})$ corresponds to the model distribution which is learned using samples from the data distribution instead. To address this issue, we derive two *predictive* estimators, one generalised from the non-parametric estimator and the other minimises KSD using parametric approximations.

**Predictions using the non-parametric estimator.** Let us consider an unseen datum $\boldsymbol{y}$. If $\boldsymbol{y}$ is sampled from $q$, then one can also apply the non-parametric estimator (9) for gradient approximation, given the observed data $\mathbf{X} = \{\boldsymbol{x}^1, ..., \boldsymbol{x}^K\} \sim q$. Concretely, if writing $\hat{\boldsymbol{g}}(\boldsymbol{y}) \approx \nabla_{\boldsymbol{y}} \log q(\boldsymbol{y}) \in \mathbb{R}^{d \times 1}$ then the non-parametric Stein gradient estimator computed on $\mathbf{X} \cup \{\boldsymbol{y}\}$ is

$$\begin{bmatrix} \hat{\boldsymbol{g}}(\boldsymbol{y})^{\text{T}} \\ \hat{\mathbf{G}} \end{bmatrix} = -(\mathbf{K}^* + \eta \boldsymbol{I})^{-1} \begin{bmatrix} \nabla_{\boldsymbol{y}} \mathcal{K}(\boldsymbol{y}, \boldsymbol{y}) + \sum_{k=1}^{K} \nabla_{\boldsymbol{x}^k} \mathcal{K}(\boldsymbol{y}, \boldsymbol{x}^k) \\ \langle \nabla, \mathbf{K} \rangle + \nabla_{\boldsymbol{y}} \mathcal{K}(\cdot, \boldsymbol{y}) \end{bmatrix}, \quad \mathbf{K}^* = \begin{bmatrix} \mathbf{K}_{\boldsymbol{y}\boldsymbol{y}} & \mathbf{K}_{\boldsymbol{y}\mathbf{X}} \\ \mathbf{K}_{\mathbf{X}\boldsymbol{y}} & \mathbf{K} \end{bmatrix},$$

with $\nabla_{\boldsymbol{y}} \mathcal{K}(\cdot, \boldsymbol{y})$ denoting a $K \times d$ matrix with rows $\nabla_{\boldsymbol{y}} \mathcal{K}(\boldsymbol{x}^k, \boldsymbol{y})$, and $\nabla_{\boldsymbol{y}} \mathcal{K}(\boldsymbol{y}, \boldsymbol{y})$ only differentiates through the second argument. Then we demonstrate in the appendix that, by simple matrix calculations and assuming a translation invariant kernel, we have (with column vector $\mathbf{1} \in \mathbb{R}^{K \times 1}$):

$$\nabla_{\boldsymbol{y}} \log q(\boldsymbol{y})^{\text{T}} \approx - \left( \mathbf{K}_{\boldsymbol{y}\boldsymbol{y}} + \eta - \mathbf{K}_{\boldsymbol{y}\mathbf{X}} (\mathbf{K} + \eta \mathbf{I})^{-1} \mathbf{K}_{\mathbf{X}\boldsymbol{y}} \right)^{-1} \\ \left( \mathbf{K}_{\boldsymbol{y}\mathbf{X}} \hat{\mathbf{G}}_V^{\text{Stein}} - \left( \mathbf{K}_{\boldsymbol{y}\mathbf{X}} (\mathbf{K} + \eta \mathbf{I})^{-1} + \mathbf{1}^{\text{T}} \right) \nabla_{\boldsymbol{y}} \mathcal{K}(\cdot, \boldsymbol{y}) \right). \tag{18}$$

In practice one would store the computed gradient $\hat{\mathbf{G}}_V^{\text{Stein}}$, the kernel matrix inverse $(\mathbf{K} + \eta \mathbf{I})^{-1}$ and $\eta$ as the "parameters" of the predictive estimator. For a new observation $\boldsymbol{y} \sim p$ in general, one can "pretend" $\boldsymbol{y}$ is a sample from $q$ and apply the above estimator as well. The approximation quality depends on the similarity between $q$ and $p$, and we conjecture here that this similarity measure, if can be described, is closely related to the KSD.

**Fitting a parametric estimator using KSD.** The non-parametric predictive estimator could be computationally demanding. Setting aside the cost of fitting the "parameters", in prediction the time complexity for the non-parametric estimator is $\mathcal{O}(K^2 + Kd)$. Also storing the "parameters" needs $\mathcal{O}(Kd)$ memory for $\hat{\mathbf{G}}_V^{\text{Stein}}$. These costs make the non-parametric estimator undesirable for high-dimensional data, since in order to obtain accurate predictions it often requires $K$ scaling with $d$ as well. To address this, one can also minimise the KSD using parametric approximations, in a similar way as to derive the score matching estimator in Section 3.3.2. More precisely, we define a parametric approximation in a similar fashion as (17), and in the appendix we show that if the RBF kernel is used for both the KSD and the parametric approximation, then the linear coefficients $\boldsymbol{a} = (a_1, ..., a_K)^{\text{T}}$ can be calculated analytically: $\hat{\boldsymbol{a}}_V^{\text{Stein}} = (\boldsymbol{\Lambda} + \eta\mathbf{I})^{-1}\boldsymbol{b}$, where

$$
\begin{aligned}
\boldsymbol{\Lambda} =& \mathbb{X} \odot (\mathbf{KKK}) + \mathbf{K}(\mathbf{K} \odot \mathbb{X})\mathbf{K} - ((\mathbf{KK}) \odot \mathbb{X})\mathbf{K} - \mathbf{K}((\mathbf{KK}) \odot \mathbb{X}), \\
\boldsymbol{b} =& (\mathbf{K}\text{diag}(\mathbb{X})\mathbf{K} + (\mathbf{KK}) \odot \mathbb{X} - \mathbf{K}(\mathbf{K} \odot \mathbb{X}) - (\mathbf{K} \odot \mathbb{X})\mathbf{K})\mathbf{1},
\end{aligned}
\tag{19}
$$

with $\mathbb{X}$ the "gram matrix" that has elements $\mathbb{X}_{ij} = (\boldsymbol{x}^i)^{\text{T}}\boldsymbol{x}^j$. Then for an unseen observation $\boldsymbol{y} \sim p$ the gradient approximation returns $\nabla_{\boldsymbol{y}} \log q(\boldsymbol{y}) \approx (\hat{\boldsymbol{a}}_V^{\text{Stein}})^{\text{T}}\nabla_{\boldsymbol{y}}\mathcal{K}(\cdot, \boldsymbol{y})$. In this case one only maintains the linear coefficients $\hat{\boldsymbol{a}}_V^{\text{Stein}}$ and computes a linear combination in prediction, which takes $\mathcal{O}(K)$ memory and $\mathcal{O}(Kd)$ time and therefore is computationally cheaper than the non-parametric prediction model (27).

# 4    Applications

We present some case studies that apply the gradient estimators to implicit models. Detailed settings (architecture, learning rate, etc.) are presented in the appendix. Implementation is released at `https://github.com/YingzhenLi/SteinGrad`.

## 4.1    Synthetic example: Hamiltonian flow with approximate gradients

We first consider a simple synthetic example to demonstrate the accuracy of the proposed gradient estimator. More precisely we consider the kernel induced Hamiltonian flow (*not* an exact sampler) (Strathmann et al., 2015) on a 2-dimensional banana-shaped object: $\boldsymbol{x} \sim \mathcal{B}(\boldsymbol{x}; b = 0.03, v = 100) \Leftrightarrow x_1 \sim \mathcal{N}(x_1; 0, v), x_2 = \epsilon + b(x_1^2 - v), \epsilon \sim \mathcal{N}(\epsilon; 0, 1)$. The approximate Hamiltonian flow is constructed using the same operator as in Hamiltonian Monte Carlo (HMC) (Neal et al., 2011), except that the exact score function $\nabla_{\boldsymbol{x}} \log \mathcal{B}(\boldsymbol{x})$ is replaced by the approximate gradients. We still use the exact target density to compute the rejection step as we mainly focus on testing the accuracy of the gradient estimators. We test both versions of the predictive Stein gradient estimator (see section 3.4) since we require the particles of parallel chains to be independent with each other. We fit the gradient estimators on $K = 200$ training datapoints from the target density. The bandwidth of the RBF kernel is computed by the median heuristic and scaled up by a scalar between $[1, 5]$. All three methods are simulated for $T = 2,000$ iterations, share the same initial locations that are constructed by target distribution samples plus Gaussian noises of standard deviation 2.0, and the results are averaged over 200 parallel chains.

We visualise the samples and some MCMC statistics in Figure 2. In general all the resulting Hamiltonian flows are HMC-like, which give us the confidence that the gradient estimators extrapolate reasonably well at unseen locations. However all of these methods have trouble exploring the extremes, because at those locations there are very few or even no training data-points. Indeed we found it necessary to use large (but not too large) bandwidths, in order to both allow exploration of those extremes, and ensure that the corresponding test function is not too smooth. In terms of quantitative metrics, the acceptance rates are reasonably high for all the gradient estimators, and the KSD estimates (across chains) as a measure of sample quality are also close to that computed on HMC samples. The returned estimates of $\mathbb{E}[x_1]$ are close to zero which is the ground true value. We found that the non-parametric Stein gradient estimator is more sensitive to hyper-parameters of the dynamics, e.g. the stepsize of each HMC step. We believe a careful selection of the kernel (e.g. those with long tails) and a better search for the hyper-parameters (for both the kernel and the dynamics) can further improve the sample quality and the chain mixing time, but this is not investigated here.

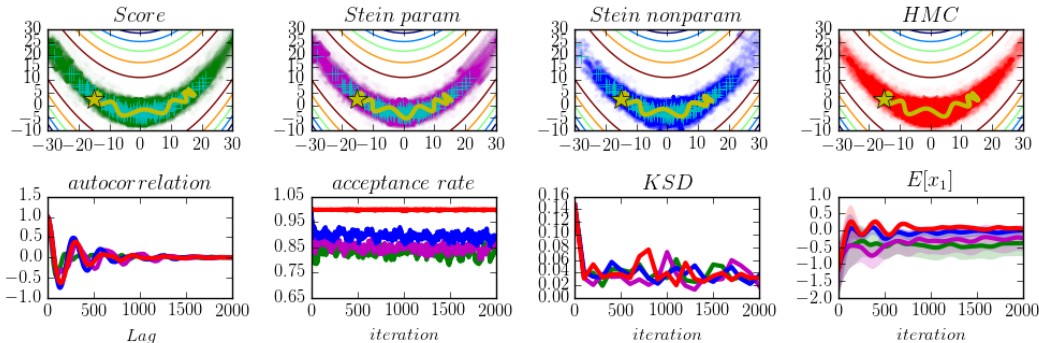

Figure 2: Kernel induced Hamiltonian flow compared with HMC. Top: samples generated from the dynamics, training data (in cyan), an the trajectory of a particle for $T = 1$ to $200$ starting at the star location (in yellow). Bottom: statistics computed during simulations. See main text for details.

## 4.2 META-LEARNING OF APPROXIMATE POSTERIOR SAMPLERS FOR BAYESIAN NNS

One of the recent focuses on meta-learning has been on learning optimisers for training deep neural networks, e.g. see (Andrychowicz et al., 2016). Could analogous goals be achieved for approximate inference? In this section we attempt to learn an approximate posterior sampler for Bayesian neural networks (Bayesian NNs, BNNs) that generalises to *unseen* datasets and architectures. A more detailed introduction of Bayesian neural networks is included in the appendix, and in a nutshell, we consider a binary classification task: $p(y = 1|\boldsymbol{x}, \boldsymbol{\theta}) = \text{sigmoid}(\text{NN}_{\boldsymbol{\theta}}(\boldsymbol{x}))$, $p_0(\boldsymbol{\theta}) = \mathcal{N}(\boldsymbol{\theta}; \mathbf{0}, \mathbf{I})$. After observing the training data $\mathcal{D} = \{(\boldsymbol{x}_n, y_n)\}_{n=1}^N$, we first obtain the approximate posterior $q_{\boldsymbol{\phi}}(\boldsymbol{\theta}) \approx p(\boldsymbol{\theta}|\mathcal{D}) \propto p_0(\boldsymbol{\theta}) \prod_{n=1}^N p(y_n|\boldsymbol{x}_n, \boldsymbol{\theta})$, then approximate the predictive distribution for a new observation as $p(y^* = 1|\boldsymbol{x}^*, \mathcal{D}) \approx \frac{1}{K} \sum_{k=1}^K p(y^* = 1|\boldsymbol{x}^*, \boldsymbol{\theta}^k), \boldsymbol{\theta}^k \sim q_{\boldsymbol{\phi}}(\boldsymbol{\theta})$. In this task we define an implicit approximate posterior distribution $q_{\boldsymbol{\phi}}(\boldsymbol{\theta})$ as the following *stochastic* normalising flow (Rezende & Mohamed, 2015) $\boldsymbol{\theta}_{t+1} = \boldsymbol{f}(\boldsymbol{\theta}_t, \nabla_t, \boldsymbol{\epsilon}_t)$: given the current location $\boldsymbol{\theta}_t$ and the mini-batch data $\{(\boldsymbol{x}_m, y_m)\}_{m=1}^M$, the update for the next step is

$$\boldsymbol{\theta}_{t+1} = \boldsymbol{\theta}_t + \zeta \Delta_{\boldsymbol{\phi}}(\boldsymbol{\theta}_t, \nabla_t) + \boldsymbol{\sigma}_{\boldsymbol{\phi}}(\boldsymbol{\theta}_t, \nabla_t) \odot \boldsymbol{\epsilon}_t, \quad \boldsymbol{\epsilon}_t \sim \mathcal{N}(\boldsymbol{\epsilon}; \mathbf{0}, \mathbf{I}),$$

$$\nabla_t = \nabla_{\boldsymbol{\theta}_t} \left[ \frac{N}{M} \sum_{m=1}^M \log p(y_m|\boldsymbol{x}_m, \boldsymbol{\theta}_t) + \log p_0(\boldsymbol{\theta}_t) \right]. \tag{20}$$

The coordinates of the noise standard deviation $\boldsymbol{\sigma}_{\boldsymbol{\phi}}(\boldsymbol{\theta}_t, \nabla_t)$ and the moving direction $\Delta_{\boldsymbol{\phi}}(\boldsymbol{\theta}_t, \nabla_t)$ are parametrised by a *coordinate-wise* neural network. If properly trained, this neural network will learn the best combination of the current location and gradient information, and produce approximate posterior samples efficiently on different probabilistic modelling tasks. Here we propose using the variational inference objective (2) computed on the samples $\{\boldsymbol{\theta}_t^k\}$ to learn the variational parameters $\boldsymbol{\phi}$. Since in this case the gradient of the log joint distribution can be computed analytically, we only approximate the gradient of the entropy term $\mathbb{H}[q]$ as in (3), with the exact score function replaced by the presented gradient estimators. We report the results using the non-parametric Stein gradient estimator as we found it works better than the parametric version. The RBF kernel is applied for gradient estimation, with the hyper-parameters determined by a grid search on the bandwidth $\sigma^2 \in \{0.25, 1.0, 4.0, 10.0, \text{median trick}\}$ and $\eta \in \{0.1, 0.5, 1.0, 2.0\}$.

We briefly describe the test protocol. We take from the UCI repository (Lichman, 2013) six binary classification datasets (australian, breast, crabs, ionosphere, pima, sonar), train an approximate sampler on crabs with a small neural network that has one 20-unit hidden layer with *ReLU* activation, and generalise to the remaining datasets with a bigger network that has 50 hidden units and uses *sigmoid* activation. We use ionosphere as the validation set to tune $\zeta$. The remaining 4 datasets are further split into 40% training subset for simulating samples from the approximate sampler, and 60% test subsets for evaluating the sampler's performance.

Figure 3 presents the (negative) test log-likelihood (LL), classification error, and an estimate of the KSD U-statistic $\mathcal{S}_U^2(p(\boldsymbol{\theta}|\mathcal{D}), q(\boldsymbol{\theta}))$ (with data sub-sampling) over 5 splits of each test dataset. Besides the gradient estimators we also compare with two baselines: an approximate posterior sampler trained by maximum a posteriori (MAP), and stochastic gradient Langevin dynamics (SGLD)

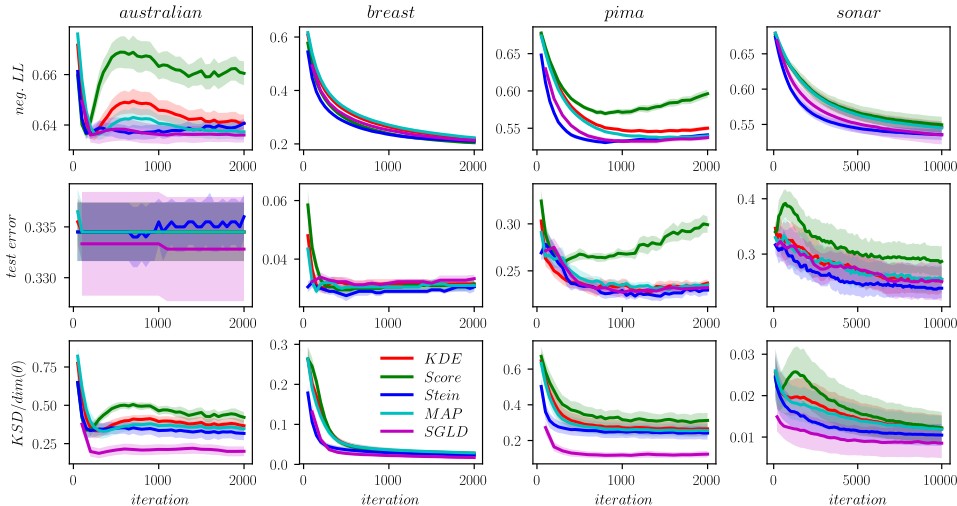

Figure 3: Generalisation performances for trained approximate posterior samplers.

(Welling & Teh, 2011) evaluated on the test datasets directly. In summary, SGLD returns best results in KSD metric. The Stein approach performs equally well or a little better than SGLD in terms of test-LL and test error. The KDE method is slightly worse and is close to MAP, indicating that the KDE estimator does not provide a very informative gradient for the entropy term. Surprisingly the score matching estimator method produces considerably worse results (except for breast dataset), even after carefully tuning the bandwidth and the regularisation parameter $\eta$. Future work should investigate the usage of advanced recurrent neural networks such as an LSTM (Hochreiter & Schmidhuber, 1997), which is expected to return better performance.

### 4.3    TOWARDS ADDRESSING MODE COLLAPSE IN GANS USING ENTROPY REGULARISATION

GANs are notoriously difficult to train in practice. Besides the instability of gradient-based minimax optimisation which has been partially addressed by many recent proposals (Salimans et al., 2016; Arjovsky et al., 2017; Berthelot et al., 2017), they also suffer from mode collapse. We propose adding an entropy regulariser to the GAN generator loss. Concretely, assume the generative model $p_{\boldsymbol{\theta}}(\boldsymbol{x})$ is implicitly defined by $\boldsymbol{x} = \boldsymbol{f}_{\boldsymbol{\theta}}(\boldsymbol{z}), \boldsymbol{z} \sim p_0(\boldsymbol{z})$, then the generator's loss is defined by

$$\tilde{\mathcal{J}}_{\text{gen}}(\boldsymbol{\theta}) = \mathcal{J}_{\text{gen}}(\boldsymbol{\theta}) - \alpha \mathbb{H}[p_{\boldsymbol{\theta}}(\boldsymbol{x})], \tag{21}$$

where $\mathcal{J}_{\text{gen}}(\boldsymbol{\theta})$ is the original loss function for the generator from any GAN algorithm and $\alpha$ is a hyper-parameter. In practice (the gradient of) (21) is estimated using Monte Carlo.

We empirically investigate the entropy regularisation idea on the very recently proposed boundary equilibrium GAN (BEGAN) (Berthelot et al., 2017) method using (continuous) MNIST, and we refer to the appendix for the detailed mathematical set-up. In this case the non-parametric V-statistic Stein gradient estimator is used. We use a convolutional generative network and a convolutional auto-encoder and select the hyper-parameters of BEGAN $\gamma \in \{0.3, 0.5, 0.7\}$, $\alpha \in [0, 1]$ and $\lambda = 0.001$. The Epanechnikov kernel $\mathcal{K}(\boldsymbol{x}, \boldsymbol{x}') := \frac{1}{d}\sum_{j=1}^{d}(1 - (x_j - x_j')^2)$ is used as the pixel values lie in a unit interval (see appendix for the expression of the score matching estimator), and to ensure the boundary condition we clip the pixel values into range $[10^{-8}, 1 - 10^{-8}]$. The generated images are visualised in Figure 4. BEGAN without the entropy regularisation fails to generate diverse samples even when trained with learning rate decay. The other three images clearly demonstrate the benefit of the entropy regularisation technique, with the Stein approach obtaining the highest diversity without compromising visual quality.

We further consider four metrics to assess the trained models quantitatively. First 500 samples are generated for each trained model, then we compute their nearest neighbours in the training set using $l_1$ distance, and obtain a probability vector $\mathbf{p}$ by averaging over these neighbour images' label vectors. In Figure 5 we depict the entropy of $\mathbf{p}$ (top left), averaged $l_1$ distances to the nearest neighbour (top right), and the difference between the largest and smallest elements in $\mathbf{p}$ (bottom right). The error bars are obtained by 5 independent runs. These results demonstrate that the Stein

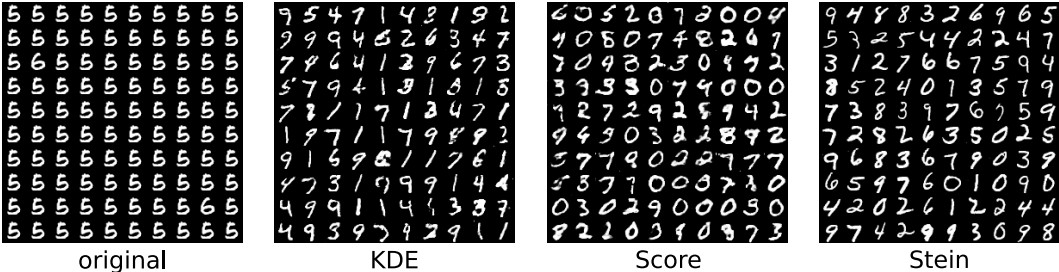

original          KDE          Score          Stein

Figure 4: Visualisation of generated images from trained BEGAN models.

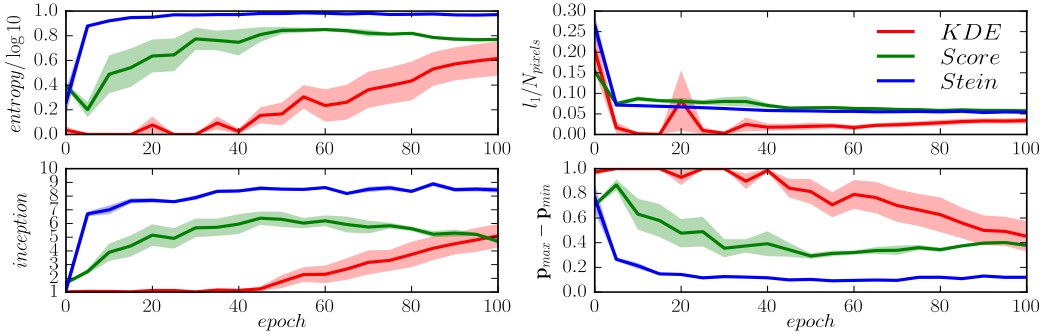

Figure 5: Quantitative evaluation on entropy regularised BEGAN. The higher the better for the LHS panels and the other way around for the RHS ones. See main text for details.

approach performs significantly better than the other two, in that it learns a better generative model not only faster but also in a more stable way. Interestingly the KDE approach achieves the lowest average $l_1$ distance to nearest neighbours, possibly because it tends to memorise training examples. We next train a fully connected network $\pi(\boldsymbol{y}|\boldsymbol{x})$ on MNIST that achieves 98.16% text accuracy, and compute on the generated images an empirical estimate of the inception score (Salimans et al., 2016) $\mathbb{E}_{p(\boldsymbol{x})}[\mathrm{KL}[\pi(\boldsymbol{y}|\boldsymbol{x})||\pi(\boldsymbol{y})]]$ with $\pi(\boldsymbol{y}) = \mathbb{E}_{p(\boldsymbol{x})}[\pi(\boldsymbol{y}|\boldsymbol{x})]$ (bottom left panel). High inception score indicates that the generate images tend to be both realistic looking and diverse, and again the Stein approach out-performs the others on this metric by a large margin.

Concerning computation speed, all the three methods are of the same order: 10.20s/epoch for KDE, 10.85s/epoch for Score, and 10.30s/epoch for Stein.[1] This is because $K < d$ (in the experiments $K = 100$ and $d = 784$) so that the complexity terms are dominated by kernel computations ($\mathcal{O}(K^2 d)$) required by all the three methods. Also for a comparison, the original BEGAN method without entropy regularisation runs for 9.05s/epoch. Therefore the main computation cost is dominated by the optimisation of the discriminator/generator, and the proposed entropy regularisation can be applied to many GAN frameworks with little computational burden.

## 5   CONCLUSIONS AND FUTURE WORK

We have presented the Stein gradient estimator as a novel generalisation to the score matching gradient estimator. With a focus on learning implicit models, we have empirically demonstrated the efficacy of the proposed estimator by showing how it opens the door to a range of novel learning tasks: approximating gradient-free MCMC, meta-learning for approximate inference, and unsupervised learning for image generation. Future work will expand the understanding of gradient estimators in both theoretical and practical aspects. Theoretical development will compare both the V-statistic and U-statistic Stein gradient estimators and formalise consistency proofs. Practical work will improve the sample efficiency of kernel estimators in high dimensions and develop fast yet accurate approximations to matrix inversion. It is also interesting to investigate applications of gradient approximation methods to training implicit generative models without the help of discriminators. Finally it remains an open question that how to generalise the Stein gradient estimator to non-kernel settings and discrete distributions.

---

[1] All the methods are timed on a machine with an NVIDIA GeForce GTX TITAN X GPU.

## ACKNOWLEDGEMENT

We thank Marton Havasi, Jiri Hron, David Janz, Qiang Liu, Maria Lomeli, Cuong Viet Nguyen and Mark Rowland for their comments and helps on the manuscript. We also acknowledge the anonymous reviewers for their review. Yingzhen Li thanks Schlumberger Foundation FFTF fellowship. Richard E. Turner thanks Google and EPSRC grants EP/M0269571 and EP/L000776/1.

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

# A  SCORE MATCHING ESTIMATOR: REMARKS AND DERIVATIONS

In this section we provide more discussions and analytical solutions for the score matching estimator. More specifically, we will derive the linear coefficient $\boldsymbol{a} = (a_1, ..., a_K)$ for the case of the Epanechnikov kernel.

## A.1  SOME REMARKS ON SCORE MATCHING

**Remark.** It has been shown in Särelä & Valpola (2005); Alain & Bengio (2014) that de-noising auto-encoders (DAEs) (Vincent et al., 2008), once trained, can be used to compute the score function approximately. Briefly speaking, a DAE learns to reconstruct a datum $\boldsymbol{x}$ from a corrupted input $\tilde{\boldsymbol{x}} = \boldsymbol{x} + \sigma\boldsymbol{\epsilon}, \boldsymbol{\epsilon} \sim \mathcal{N}(\boldsymbol{0}, \mathbf{I})$ by minimising the mean square error. Then the optimal DAE can be used to approximate the score function as $\nabla_{\boldsymbol{x}} \log p(\boldsymbol{x}) \approx \frac{1}{\sigma^2}(\text{DAE}^*(\boldsymbol{x}) - \boldsymbol{x})$. Sonderby et al. (2017) applied this idea to train an implicit model for image super-resolution, providing some promising results in some metrics. However applying similar ideas to variational inference can be computationally expensive, because the estimation of $\nabla_{\boldsymbol{z}} \log q(\boldsymbol{z}|\boldsymbol{x})$ is a sub-routine for VI which is repeatedly required. Therefore in the paper we deploy kernel machines that allow analytical solutions to the score matching estimator in order to avoid double loop optimisation.

**Remark.** As a side note, score matching can also be used to learn the parameters of an unnormalised density. In this case the target distribution $q$ would be the data distribution and $\hat{q}$ is often a Boltzmann distribution with intractable partition function. As a parameter estimation technique, score matching is also related to contrastive divergence (Hinton, 2002), pseudo likelihood estimation (Hyvärinen, 2006), and DAEs (Vincent, 2011; Alain & Bengio, 2014). Generalisations of score matching methods are also presented in e.g. Lyu (2009); Marlin et al. (2010).

## A.2  THE RBF KERNEL CASE

The derivations for the RBF kernel case is referred to (Strathmann et al., 2015), and for completeness we include the final solutions here. Assume the parametric approximation is defined as $\log \hat{q}(\boldsymbol{x}) = \sum_{k=1}^{K} a_k \mathcal{K}(\boldsymbol{x}, \boldsymbol{x}^k) + C$, where the RBF kernel uses bandwidth parameter $\sigma$. then the optimal solution of the coefficients $\hat{\boldsymbol{a}}^{\text{score}} = (\boldsymbol{\Sigma} + \eta\mathbf{I})^{-1}\boldsymbol{v}$, with

$$\boldsymbol{v} = \sum_{i=1}^{d} \left[ \sigma^2 \mathbf{K}\mathbf{1} - \left( \mathbf{K}(\mathbf{x}_i \odot \mathbf{x}_i) + \text{diag}(\mathbf{x}_i)\mathbf{K}\mathbf{1} - 2\text{diag}(\mathbf{x}_i)\mathbf{K}\mathbf{x}_i \right) \right],$$

$$\boldsymbol{\Sigma} = \sum_{i=1}^{d} [\text{diag}(\mathbf{x}_i)\mathbf{K} - \mathbf{K}\text{diag}(\mathbf{x}_i)] [\mathbf{K}\text{diag}(\mathbf{x}_i) - \text{diag}(\mathbf{x}_i)\mathbf{K}],$$

$$\mathbf{x}_i = (x_i^1, x_i^2, ..., x_i^K)^{\mathrm{T}} \in \mathbb{R}^{K \times 1}.$$

## A.3  THE EPANECHNIKOV KERNEL CASE

The Epanechnikov kernel is defined as $\mathcal{K}(\boldsymbol{x}, \boldsymbol{x}') = \frac{1}{d} \sum_{i=1}^{d} (1 - (x_i - x_i')^2)$, where the first and second order gradients w.r.t. $x_i$ is

$$\nabla_{x_i} \mathcal{K}(\boldsymbol{x}, \boldsymbol{x}') = \frac{2}{d}(x_i' - x_i), \quad \nabla_{x_i} \nabla_{x_i} \mathcal{K}(\boldsymbol{x}, \boldsymbol{x}') = -\frac{2}{d}. \tag{22}$$

Thus the score matching objective with $\log \hat{q}(\boldsymbol{x}) = \sum_{k=1}^{K} a_k \mathcal{K}(\boldsymbol{x}, \boldsymbol{x}^k) + C$ is reduced to

$$\mathcal{F}(\boldsymbol{a}) = \frac{1}{K} \sum_{j=1}^{K} \left[ ||\sum_{k=1}^{K} a_k \frac{2}{d}(\boldsymbol{x}^k - \boldsymbol{x}^j)||_2^2 - 2 \sum_{k=1}^{K} a_k \frac{2}{d} d \right]$$

$$= \frac{4}{K} \sum_{j=1}^{K} \left[ \frac{1}{d^2} \sum_{k=1}^{K} \sum_{k'=1}^{K} a_k a_{k'} (\boldsymbol{x}^k - \boldsymbol{x}^j)^{\mathrm{T}} (\boldsymbol{x}^{k'} - \boldsymbol{x}^j) - \boldsymbol{a}^{\mathrm{T}}\mathbf{1} \right]$$

$$:= 4(\boldsymbol{a}^{\mathrm{T}}\boldsymbol{\Sigma}\boldsymbol{a} - \boldsymbol{a}^{\mathrm{T}}\mathbf{1}),$$

with the matrix elements

$$\mathbf{\Sigma}_{kk'} = \frac{1}{d^2} \left[ (\boldsymbol{x}^k)^{\mathrm{T}} \boldsymbol{x}^{k'} + \frac{1}{K} \sum_{j=1}^{K} \left( ||\boldsymbol{x}^j||_2^2 - (\boldsymbol{x}^k + \boldsymbol{x}^{k'})^{\mathrm{T}} \boldsymbol{x}^j \right) \right].$$

Define the "gram matrix" $\mathbb{X}_{ij} = (\boldsymbol{x}^i)^{\mathrm{T}} \boldsymbol{x}^j$, we write the matrix form of $\mathbf{\Sigma}$ as

$$\mathbf{\Sigma} = \frac{1}{d^2} \left[ \mathbb{X} + \frac{1}{K} \left( \mathrm{Tr}(\mathbb{X}) - 2\mathbb{X}\mathbf{1}\mathbf{1}^{\mathrm{T}} \right) \right].$$

Thus with an $l_2$ regulariser, the fitted coefficients are

$$\hat{\boldsymbol{a}}^{\mathrm{score}} = \frac{d^2}{2} \left[ \mathbb{X} + \frac{1}{K} \left( \mathrm{Tr}(\mathbb{X}) - 2\mathbb{X}\mathbf{1}\mathbf{1}^{\mathrm{T}} \right) + \eta \mathbf{I} \right]^{-1} \mathbf{1}.$$

## B STEIN GRADIENT ESTIMATOR: DERIVATIONS

### B.1 DIRECT MINIMISATION OF KSD V-STATISTIC AND U-STATISTIC

The V-statistic of KSD is the following: given samples $\boldsymbol{x}^k \sim q, k = 1, ..., K$ and recall $\mathbf{K}_{jl} = \mathcal{K}(\boldsymbol{x}^j, \boldsymbol{x}^l)$

$$\mathcal{S}_V^2(q, \hat{q}) = \frac{1}{K^2} \sum_{j=1}^{K} \sum_{l=1}^{K} \left[ \hat{\boldsymbol{g}}(\boldsymbol{x}^j)^{\mathrm{T}} \mathbf{K}_{jl} \hat{\boldsymbol{g}}(\boldsymbol{x}^l) + \hat{\boldsymbol{g}}(\boldsymbol{x}^j)^{\mathrm{T}} \nabla_{\boldsymbol{x}^l} \mathbf{K}_{jl} + \nabla_{\boldsymbol{x}^j} \mathbf{K}_{jl}^{\mathrm{T}} \hat{\boldsymbol{g}}(\boldsymbol{x}^l) + \mathrm{Tr}(\nabla_{\boldsymbol{x}^j, \boldsymbol{x}^l} \mathbf{K}_{jl}) \right].$$

$$(23)$$

The last term $\nabla_{\boldsymbol{x}^j, \boldsymbol{x}^l} \mathbf{K}_{jl}$ will be ignored as it does not depend on the approximation $\hat{\boldsymbol{g}}$. Using matrix notations defined in the main text, readers can verify that the V-statistic can be computed as

$$\mathcal{S}_V^2(q, \hat{q}) = \frac{1}{K^2} \mathrm{Tr}(\mathbf{K}\hat{\mathbf{G}}\hat{\mathbf{G}}^{\mathrm{T}} + 2\langle \nabla, \mathbf{K} \rangle \hat{\mathbf{G}}^{\mathrm{T}}) + C. \tag{24}$$

Using the cyclic invariance of matrix trace leads to the desired result in the main text. The U-statistic of KSD removes terms indexed by $j = l$ in (23), in which the matrix form is

$$\mathcal{S}_U^2(q, \hat{q}) = \frac{1}{K(K-1)} \mathrm{Tr}((\mathbf{K} - \mathrm{diag}(\mathbf{K}))\hat{\mathbf{G}}\hat{\mathbf{G}}^{\mathrm{T}} + 2(\langle \nabla, \mathbf{K} \rangle - \nabla\mathrm{diag}(\mathbf{K}))\hat{\mathbf{G}}^{\mathrm{T}}) + C. \tag{25}$$

with the $j$th row of $\nabla\mathrm{diag}(\mathbf{K})$ defined as $\nabla_{\boldsymbol{x}^j} \mathcal{K}(\boldsymbol{x}^j, \boldsymbol{x}^j)$. For most translation invariant kernels this extra term $\nabla\mathrm{diag}(\mathbf{K}) = \mathbf{0}$, thus the optimal solution of $\hat{\mathbf{G}}$ by minimising KSD U-statistic is

$$\hat{\mathbf{G}}_U^{\mathrm{Stein}} = -(\mathbf{K} - \mathrm{diag}(\mathbf{K}) + \eta \boldsymbol{I})^{-1} \langle \nabla, \mathbf{K} \rangle. \tag{26}$$

### B.2 DERIVING THE NON-PARAMETRIC PREDICTIVE ESTIMATOR

Let us consider an unseen datum $\boldsymbol{y}$. If $\boldsymbol{y}$ is sampled from the $q$ distribution, then one can also apply the non-parametric estimator (9) for gradient approximations, given the observed data $\mathbf{X} = \{\boldsymbol{x}^1, ..., \boldsymbol{x}^K\} \sim q$. Concretely, if writing $\hat{\boldsymbol{g}}(\boldsymbol{y}) \approx \nabla_{\boldsymbol{y}} \log q(\boldsymbol{y}) \in \mathbb{R}^{d \times 1}$ then the non-parametric Stein gradient estimator (using V-statistic) is

$$\begin{bmatrix} \hat{\boldsymbol{g}}(\boldsymbol{y})^{\mathrm{T}} \\ \hat{\mathbf{G}} \end{bmatrix} = -(\mathbf{K}^* + \eta \boldsymbol{I})^{-1} \begin{bmatrix} \nabla_{\boldsymbol{y}} \mathcal{K}(\boldsymbol{y}, \boldsymbol{y}) + \sum_{k=1}^{K} \nabla_{\boldsymbol{x}^k} \mathcal{K}(\boldsymbol{y}, \boldsymbol{x}^k) \\ \langle \nabla, \mathbf{K} \rangle + \nabla_{\boldsymbol{y}} \mathcal{K}(\cdot, \boldsymbol{y}) \end{bmatrix}, \quad \mathbf{K}^* = \begin{bmatrix} \mathbf{K}_{\boldsymbol{y}\boldsymbol{y}} & \mathbf{K}_{\boldsymbol{y}\mathbf{X}} \\ \mathbf{K}_{\mathbf{X}\boldsymbol{y}} & \mathbf{K} \end{bmatrix},$$

with $\nabla_{\boldsymbol{y}} \mathcal{K}(\cdot, \boldsymbol{y})$ denoting a $K \times d$ matrix with rows $\nabla_{\boldsymbol{y}} \mathcal{K}(\boldsymbol{x}^k, \boldsymbol{y})$, and $\nabla_{\boldsymbol{y}} \mathcal{K}(\boldsymbol{y}, \boldsymbol{y})$ only differentiates through the second argument. Thus by simple matrix calculations, we have:

$$\nabla_{\boldsymbol{y}} \log q(\boldsymbol{y})^{\mathrm{T}} \approx - \left( \mathbf{K}_{\boldsymbol{y}\boldsymbol{y}} + \eta - \mathbf{K}_{\boldsymbol{y}\mathbf{X}}(\mathbf{K} + \eta \mathbf{I})^{-1} \mathbf{K}_{\mathbf{X}\boldsymbol{y}} \right)^{-1}$$
$$\left( \nabla_{\boldsymbol{y}} \mathcal{K}(\boldsymbol{y}, \boldsymbol{y}) + \sum_{k=1}^{K} \nabla_{\boldsymbol{x}^k} \mathcal{K}(\boldsymbol{y}, \boldsymbol{x}^k) + \mathbf{K}_{\boldsymbol{y}\mathbf{X}} \hat{\mathbf{G}}_V^{\mathrm{Stein}} - \mathbf{K}_{\boldsymbol{y}\mathbf{X}}(\mathbf{K} + \eta \mathbf{I})^{-1} \nabla_{\boldsymbol{y}} \mathcal{K}(\cdot, \boldsymbol{y}) \right).$$
$$(27)$$

For translation invariant kernels, typically $\nabla_{\boldsymbol{y}}\mathcal{K}(\boldsymbol{y}, \boldsymbol{y}) = \boldsymbol{0}$, and more conveniently,

$$\nabla_{\boldsymbol{x}^k}\mathcal{K}(\boldsymbol{y}, \boldsymbol{x}^k) = \nabla_{\boldsymbol{x}^k}(\boldsymbol{x}^k - \boldsymbol{y})\nabla_{(\boldsymbol{x}^k-\boldsymbol{y})}\mathcal{K}(\boldsymbol{x}^k - \boldsymbol{y}) = -\nabla_{\boldsymbol{y}}\mathcal{K}(\boldsymbol{x}^k, \boldsymbol{y}).$$

Thus equation (27) can be further simplified to (with column vector $\boldsymbol{1} \in \mathbb{R}^{K \times 1}$)

$$\begin{aligned}
\nabla_{\boldsymbol{y}} \log q(\boldsymbol{y})^{\mathrm{T}} \approx &- \left(\mathbf{K}_{\boldsymbol{yy}} + \eta - \mathbf{K}_{\boldsymbol{y}\mathbf{X}}(\mathbf{K} + \eta\mathbf{I})^{-1}\mathbf{K}_{\mathbf{X}\boldsymbol{y}}\right)^{-1} \\
&\left(\mathbf{K}_{\boldsymbol{y}\mathbf{X}}\hat{\mathbf{G}}_V^{\mathrm{Stein}} - \left(\mathbf{K}_{\boldsymbol{y}\mathbf{X}}(\mathbf{K} + \eta\mathbf{I})^{-1} + \boldsymbol{1}^{\mathrm{T}}\right)\nabla_{\boldsymbol{y}}\mathcal{K}(\cdot, \boldsymbol{y})\right).
\end{aligned} \tag{28}$$

The solution for the U-statistic case can be derived accordingly which we omit here.

### B.3 PARAMETRIC STEIN GRADIENT ESTIMATOR WITH THE RBF KERNEL

We define a parametric approximation in a similar way as for the score matching estimator:

$$\log \hat{q}(\boldsymbol{x}) := \sum_{k=1}^K a_k\mathcal{K}(\boldsymbol{x}, \boldsymbol{x}^k) + C, \quad \mathcal{K}(\boldsymbol{x}, \boldsymbol{x}') = \exp\left[-\frac{1}{2\sigma^2}||\boldsymbol{x} - \boldsymbol{x}'||_2^2\right]. \tag{29}$$

Now we show the optimal solution of $\boldsymbol{a} = (a_1, ..., a_K)^{\mathrm{T}}$ by minimising (23). To simplify derivations we assume the approximation and KSD use the same kernel. First note that the gradient of the RBF kernel is

$$\nabla_{\boldsymbol{x}}\mathcal{K}(\boldsymbol{x}, \boldsymbol{x}') = \frac{1}{\sigma^2}\mathcal{K}(\boldsymbol{x}, \boldsymbol{x}')(\boldsymbol{x}' - \boldsymbol{x}). \tag{30}$$

Substituting (30) into (23):

$$\mathcal{S}_V^2(q, \hat{q}) = C + \clubsuit + 2\spadesuit,$$

$$\clubsuit = \frac{1}{K^2}\sum_{k=1}^K\sum_{k'=1}^K\sum_{j=1}^K\sum_{l=1}^K a_k a_{k'}\mathbf{K}_{kj}\mathbf{K}_{jl}\mathbf{K}_{lk'}\frac{1}{\sigma^4}(\boldsymbol{x}^k - \boldsymbol{x}^j)^{\mathrm{T}}(\boldsymbol{x}^{k'} - \boldsymbol{x}^l), \tag{31}$$

$$\spadesuit = \frac{1}{K^2}\sum_{k=1}^K\sum_{j=1}^K\sum_{l=1}^K a_k\mathbf{K}_{kj}\mathbf{K}_{jl}\frac{1}{\sigma^4}(\boldsymbol{x}^k - \boldsymbol{x}^j)^{\mathrm{T}}(\boldsymbol{x}^j - \boldsymbol{x}^l). \tag{32}$$

We first consider summing the $j, l$ indices in $\clubsuit$. Recall the "gram matrix" $\mathbb{X}_{ij} = (\boldsymbol{x}^i)^{\mathrm{T}}\boldsymbol{x}^j$, the inner product term in $\clubsuit$ can be expressed as $\mathbb{X}_{kk'} + \mathbb{X}_{jl} - \mathbb{X}_{kl} - \mathbb{X}_{jk'}$. Thus the summation over $j, l$ can be re-written as

$$\begin{aligned}
\boldsymbol{\Lambda} := &\sum_{j=1}^K\sum_{l=1}^K \mathbf{K}_{kj}\mathbf{K}_{jl}\mathbf{K}_{lk'}(\mathbb{X}_{kk'} + \mathbb{X}_{jl} - \mathbb{X}_{kl} - \mathbb{X}_{jk'}) \\
&= \mathbb{X} \odot (\mathbf{KKK}) + \mathbf{K}(\mathbf{K} \odot \mathbb{X})\mathbf{K} - ((\mathbf{KK}) \odot \mathbb{X})\mathbf{K} - \mathbf{K}((\mathbf{KK}) \odot \mathbb{X}).
\end{aligned}$$

And thus $\clubsuit = \frac{1}{\sigma^4}\boldsymbol{a}^{\mathrm{T}}\boldsymbol{\Lambda}\boldsymbol{a}$. Similarly the summation over $j, l$ in $\spadesuit$ can be simplified into

$$\begin{aligned}
-\boldsymbol{b} := &\sum_{j=1}^K\sum_{l=1}^K \mathbf{K}_{kj}\mathbf{K}_{jl}(\mathbb{X}_{kj} + \mathbb{X}_{jl} - \mathbb{X}_{kl} - \mathbb{X}_{jj}) \\
&= -(\mathbf{K}\mathrm{diag}(\mathbb{X})\mathbf{K} + (\mathbf{KK}) \odot \mathbb{X} - \mathbf{K}(\mathbf{K} \odot \mathbb{X}) - (\mathbf{K} \odot \mathbb{X})\mathbf{K})\boldsymbol{1},
\end{aligned}$$

which leads to $\spadesuit = -\frac{1}{\sigma^4}\boldsymbol{a}^{\mathrm{T}}\boldsymbol{b}$. Thus minimising $\mathcal{S}_V^2(q, \hat{q})$ plus an $l_2$ regulariser returns the Stein estimator $\hat{\boldsymbol{a}}_V^{\mathrm{Stein}}$ in the main text.

Similarly we can derive the solution for KSD U-statistic minimisation. The U statistic can also be represented in quadratic form $\mathcal{S}_U^2(q, \hat{q}) = C + \tilde{\clubsuit} + 2\tilde{\spadesuit}$, with $\tilde{\spadesuit} = \spadesuit$ and

$$\tilde{\clubsuit} = \clubsuit - \frac{1}{K^2}\sum_{k=1}^K\sum_{k'=1}^K\sum_{j=1}^K a_k a_{k'}\mathbf{K}_{kj}\mathbf{K}_{jj}\mathbf{K}_{jk'}\frac{1}{\sigma^4}(\mathbb{X}_{kk'} + \mathbb{X}_{jj} - \mathbb{X}_{kj} - \mathbb{X}_{jk'}).$$

Summing over the $j$ indices for the second term, we have

$$\sum_{j=1}^{K} \mathbf{K}_{kj}\mathbf{K}_{jj}\mathbf{K}_{jk'}(\mathbb{X}_{kk'} + \mathbb{X}_{jj} - \mathbb{X}_{kj} - \mathbb{X}_{jk'})$$

$$= \mathbb{X} \odot (\mathbf{K}\mathrm{diag}(\mathbf{K})\mathbf{K}) + \mathbf{K}\mathrm{diag}(\mathbf{K} \odot \mathbb{X})\mathbf{K} - ((\mathbf{K}\mathrm{diag}(\mathbf{K})) \odot \mathbb{X})\mathbf{K} - \mathbf{K}((\mathrm{diag}(\mathbf{K})\mathbf{K}) \odot \mathbb{X}).$$

Working through the analogous derivations reveals that $\hat{\boldsymbol{a}}_U^{\mathrm{Stein}} = (\tilde{\boldsymbol{\Lambda}} + \eta\mathbf{I})^{-1}\boldsymbol{b}$, with

$$\tilde{\boldsymbol{\Lambda}} = \mathbb{X} \odot (\mathbf{K}(\mathbf{K} - \mathrm{diag}(\mathbf{K}))\mathbf{K}) + \mathbf{K}((\mathbf{K} \odot \mathbb{X}) - \mathrm{diag}(\mathbf{K} \odot \mathbb{X}))\mathbf{K}$$
$$- ((\mathbf{K}(\mathbf{K} - \mathrm{diag}(\mathbf{K}))) \odot \mathbb{X})\mathbf{K} - \mathbf{K}(((\mathbf{K} - \mathrm{diag}(\mathbf{K}))\mathbf{K}) \odot \mathbb{X}).$$

## C MORE DETAILS ON THE EXPERIMENTS

We describe the detailed experimental set-up in this section. All experiments use Adam optimiser (Kingma & Ba, 2015) with standard parameter settings.

### C.1 APPROXIMATE POSTERIOR SAMPLER EXPERIMENTS

We start by reviewing Bayesian neural networks with binary classification as a running example. In this task, a normal deep neural network is constructed to predict $y = \boldsymbol{f_\theta}(\boldsymbol{x})$, and the neural network is parameterised by a set of weights (and bias vectors which we omit here for simplicity) $\boldsymbol{\theta} = \{\mathbf{W}^l\}_{l=1}^{L}$. In the Bayesian framework these network weights are treated as random variables, and a prior distribution, e.g. Gaussian, is also attached to them: $p_0(\boldsymbol{\theta}) = \mathcal{N}(\boldsymbol{\theta}; \mathbf{0}, \mathbf{I})$. The likelihood function of $\boldsymbol{\theta}$ is then defined as

$$p(y = 1|\boldsymbol{x}, \boldsymbol{\theta}) = \mathrm{sigmoid}(\mathrm{NN}_{\boldsymbol{\theta}}(\boldsymbol{x})),$$

and $p(y = 0|\boldsymbol{x}, \boldsymbol{\theta}) = 1 - p(y = 1|\boldsymbol{x}, \boldsymbol{\theta})$ accordingly. One can show that the usage of Bernoulli distribution here corresponds to applying cross entropy loss for training.

After framing the deep neural network as a probabilistic model, a Bayesian approach would find the posterior of the network weights $p(\boldsymbol{\theta}|\mathcal{D})$ and use the uncertainty information encoded in it for future predictions. By Bayes' rule, the exact posterior is

$$p(\boldsymbol{\theta}|\mathcal{D}) \propto p_0(\boldsymbol{\theta}) \prod_{n=1}^{N} p(y_n|\boldsymbol{x}_n, \boldsymbol{\theta}),$$

and the predictive distribution for a new input $\boldsymbol{x}^*$ is

$$p(y^* = 1|\boldsymbol{x}^*, \mathcal{D}) = \int p(y^* = 1|\boldsymbol{x}^*, \boldsymbol{\theta})p(\boldsymbol{\theta}|\mathcal{D})d\boldsymbol{\theta}. \tag{33}$$

Again the exact posterior is intractable, and approximate inference would fit an approximate posterior distribution $q_{\boldsymbol{\phi}}(\boldsymbol{\theta})$ parameterised by the variational parameters $\boldsymbol{\phi}$ to the exact posterior, and then use it to compute the (approximate) predictive distribution.

$$p(y^* = 1|\boldsymbol{x}^*, \mathcal{D}) \approx \int p(y^* = 1|\boldsymbol{x}^*, \boldsymbol{\theta})q_{\boldsymbol{\phi}}(\boldsymbol{\theta})d\boldsymbol{\theta}.$$

Since in practice analytical integration for neural network weights is also intractable, the predictive distribution is further approximated by Monte Carlo:

$$p(y^* = 1|\boldsymbol{x}^*, \mathcal{D}) \approx \frac{1}{K} \sum_{k=1}^{K} p(y^* = 1|\boldsymbol{x}^*, \boldsymbol{\theta}^k), \quad \boldsymbol{\theta}^k \sim q_{\boldsymbol{\phi}}(\boldsymbol{\theta}).$$

Now it remains to fit the approximate posterior $q_{\boldsymbol{\phi}}(\boldsymbol{\theta})$, and in the experiment the approximate posterior is implicitly constructed by a stochastic flow. For the training task, we use a one hidden layer neural network with 20 hidden units to compute the noise variance and the moving direction of the next update. In a nutshell it takes the $i$th coordinate of the current position and the gradient $\boldsymbol{\theta}_t(i), \nabla_t(i)$ as the inputs, and output the corresponding coordinate of the moving direction

$\Delta_{\boldsymbol{\phi}}(\boldsymbol{\theta}_t, \nabla_t)(i)$ and the noise variance $\boldsymbol{\sigma}_{\boldsymbol{\phi}}(\boldsymbol{\theta}_t, \nabla_t)(i)$. Softplus non-linearity is used for the hidden layer and to compute the noise variance we apply ReLU activation to ensure non-negativity. The step-size $\zeta$ is selected as 1e-5 which is tuned on the KDE approach. For SGLD step-size 1e-5 also returns overall good results.

The training process is the following. We simulate the approximate sampler for 10 transitions and sum over the variational lower-bounds computed on the samples of every step. Concretely, the maximisation objective is

$$\mathcal{L}(\boldsymbol{\phi}) = \sum_{t=1}^{T} \mathcal{L}_{\text{VI}}(q_t),$$

where $T = 100$ and $q_t(\boldsymbol{\theta})$ is implicitly defined by the marginal distribution of $\boldsymbol{\theta}_t$ that is dependent on $\boldsymbol{\phi}$. In practice the variational lower-bound $\mathcal{L}_{\text{VI}}(q_t)$ is further approximated by Monte Carlo and data sub-sampling:

$$\mathcal{L}_{\text{VI}}(q_t) \approx \frac{N}{M} \sum_{m=1}^{M} \log p(y_m | \boldsymbol{x}_m, \boldsymbol{\theta}_t) + \log p_0(\boldsymbol{\theta}_t) - \log q_t(\boldsymbol{\theta}_t).$$

The MAP baseline considers an alternative objective function by removing the $\log q_t(\boldsymbol{\theta}_t)$ term from the above MC-VI objective.

Truncated back-propagation is applied for every 10 steps in order to avoid vanishing/exploding gradients. The simulated samples at time $T$ are stored to initialise the Markov chain for the next iteration, and for every 50 iterations we restart the simulation by randomly sampling the locations from the prior. Early stopping is applied using the validation dataset, and the learning rate is set to 0.001, the number of epochs is set to 500.

We perform hyper-parameter search for the kernel, i.e. a grid search on the bandwidth $\sigma^2 \in \{0.25, 1.0, 4.0, 10.0, \text{median trick}\}$ and $\eta \in \{0.1, 0.5, 1.0, 2.0\}$. We found the median heuristic is sufficient for the KDE and Stein approaches. However, we failed to obtain desirable results using the score matching estimator with median heuristics, and for other settings the score matching approach underperforms when compared to KDE and Stein methods.

## C.2 BEGAN EXPERIMENTS

In this section we describe the experimental details of the BEGAN experiment, but first we introduce the mathematical idea and discuss how the entropy regulariser is applied.

Assume the generator is implicitly defined: $\boldsymbol{x} \sim p_{\boldsymbol{\theta}}(\boldsymbol{x}) \leftrightarrow \boldsymbol{x} = \boldsymbol{f}_{\boldsymbol{\theta}}(\boldsymbol{z}), \boldsymbol{z} \sim p_0(\boldsymbol{z})$. In BEGAN the discriminator is defined as an auto-encoder $D_{\boldsymbol{\varphi}}(\boldsymbol{x})$ that reconstructs the input $\boldsymbol{x}$. After selecting a ratio parameter $\gamma > 0$, a control rate $\beta_0$ initialised at 0, and a "learning rate" $\lambda > 0$ for the control rate, the loss functions for the generator $\boldsymbol{x} = \boldsymbol{f}_{\boldsymbol{\theta}}(\boldsymbol{z}), \boldsymbol{z} \sim p_0(\boldsymbol{z})$ and the discriminator are:

$$
\begin{aligned}
\mathcal{J}(\boldsymbol{x}) &= ||D_{\boldsymbol{\varphi}}(\boldsymbol{x}) - \boldsymbol{x}||, \quad ||\cdot|| = ||\cdot||_2^2 \text{ or } ||\cdot||_1, \\
\mathcal{J}_{\text{gen}}(\boldsymbol{\theta}; \boldsymbol{\varphi}) &= \mathcal{J}(\boldsymbol{f}_{\boldsymbol{\theta}}(\boldsymbol{z})), \quad \boldsymbol{z} \sim p_0(\boldsymbol{z}) \\
\mathcal{J}_{\text{dis}}(\boldsymbol{\varphi}; \boldsymbol{\theta}) &= \mathcal{J}(\boldsymbol{x}) - \beta_t \mathcal{J}_{\text{gen}}(\boldsymbol{\theta}; \boldsymbol{\varphi}), \quad \boldsymbol{x} \sim \mathcal{D} \\
\beta_{t+1} &= \beta_t + \lambda(\gamma \mathcal{J}(\boldsymbol{x}) - \mathcal{J}(\boldsymbol{f}_{\boldsymbol{\theta}}(\boldsymbol{z}))).
\end{aligned}
\tag{34}
$$

The main idea behind BEGAN is that, as the reconstruction loss $\mathcal{J}(\cdot)$ is approximately Gaussian distributed, with $\gamma = 1$ the discriminator loss $\mathcal{J}_{\text{dis}}$ is (approximately) proportional to the Wasserstein distance between loss distributions induced by the data distribution $p_{\mathcal{D}}(\boldsymbol{x})$ and the generator $p_{\boldsymbol{\theta}}(\boldsymbol{x})$. In practice it is beneficial to maintain the equilibrium $\gamma \mathbb{E}_{p_{\mathcal{D}}}[\mathcal{J}(\boldsymbol{x})] = \mathbb{E}_{p_{\boldsymbol{\theta}}}[\mathcal{J}(\boldsymbol{x})]$ through the optimisation procedure described in (34) that is motivated by proportional control theory. This approach effectively stabilises training, however it suffers from catastrophic mode collapsing problem (see the left most panel in Figure 4). To address this issue, we simply subtract an entropy term from the generator's loss function, i.e.

$$\tilde{\mathcal{J}}_{\text{gen}}(\boldsymbol{\theta}; \boldsymbol{\varphi}) = \mathcal{J}_{\text{gen}}(\boldsymbol{\theta}; \boldsymbol{\varphi}) - \alpha \mathbb{H}[p_{\boldsymbol{\theta}}], \tag{35}$$

where the rest of the optimisation objectives remains as in (34). This procedure would maintain the equilibrium $\gamma \mathbb{E}_{p_{\mathcal{D}}}[\mathcal{J}(\boldsymbol{x})] = \mathbb{E}_{p_{\boldsymbol{\theta}}}[\mathcal{J}(\boldsymbol{x})] - \alpha \mathbb{H}[p]$. We approximate the gradient $\nabla_{\boldsymbol{\theta}} \mathbb{H}[p_{\boldsymbol{\theta}}]$ using the estimators presented in the main text. For the purpose of updating the control rate $\beta_t$ two

strategies are considered to approximate the contribution of the entropy term. Given $K$ samples $\boldsymbol{x}^1, ..., \boldsymbol{x}^k \sim p_{\boldsymbol{\theta}}(\boldsymbol{x})$, The first proposal considers a plug-in estimate of the entropy term with a KDE estimate of $p_{\boldsymbol{\theta}}(\boldsymbol{x})$, which is consistent with the KDE estimator but not necessary with the other two (as they use kernels when representing $\log p_{\boldsymbol{\theta}}(\boldsymbol{x})$ or $\nabla_{\boldsymbol{x}} \log p_{\boldsymbol{\theta}}(\boldsymbol{x})$). The second one uses a proxy of the entropy loss $-\tilde{\mathbb{H}}[p] \approx \frac{1}{K} \sum_{k=1}^{K} \nabla_{\boldsymbol{x}^k} \log p_{\boldsymbol{\theta}}(\boldsymbol{x}^k)^{\mathrm{T}} \boldsymbol{x}^k$ with generated samples $\{\boldsymbol{x}^k\}$ and $\nabla_{\boldsymbol{x}^k} \log p_{\boldsymbol{\theta}}(\boldsymbol{x}^k)$ approximated by the gradient estimator in use.

In the experiment, we construct a deconvolutional net for the generator and a convolutional auto-encoder for the discriminator. The convolutional encoder consists of 3 convolutional layers with filter width 3, stride 2, and number of feature maps [32, 64, 64]. These convolutional layers are followed by two fully connected layers with [512, 64] units. The decoder and the generative net have a symmetric architecture but with stride convolutions replaced by deconvolutions. ReLU activation function is used for all layers except the last layer of the generator, which uses sigmoid non-linearity. The reconstruction loss in use is the squared $\ell_2$ norm $|| \cdot ||_2^2$. The randomness $p_0(\boldsymbol{z})$ is selected as uniform distribution in [-1, 1] as suggested in the original paper (Berthelot et al., 2017). The mini-batch size is set to $K = 100$. Learning rate is initialised at 0.0002 and decayed by 0.9 every 10 epochs, which is tuned on the KDE model. The selected $\gamma$ and $\alpha$ values are: for KDE estimator approach $\gamma = 0.3, \alpha\gamma = 0.05$, for score matching estimator approach $\gamma = 0.3, \alpha\gamma = 0.1$, and for Stein approach $\gamma = 0.5$ and $\alpha\gamma = 0.3$. The presented results use the KDE plug-in estimator for the entropy estimates (used to tune $\beta$) for the KDE and score matching approaches. Initial experiments found that for the Stein approach, using the KDE entropy estimator works slightly worse than the proxy loss, thus we report results using the proxy loss. An advantage of using the proxy loss is that it directly relates to the approximate gradient. Furthermore we empirically observe that the performance of the Stein approach is much more robust to the selection of $\gamma$ and $\alpha$ when compared to the other two methods.

