# OpenReview forum: "Gradient Estimators for Implicit Models"
_ICLR.cc/2018/Conference — Accept (Poster)_

### Official Review · AnonReviewer2 · 2017-11-26
**Interesting idea**

**Rating:** 7
**Confidence:** 2

**Review:**

Post rebuttal phase (see below for original comments)
================================================================================
I thank the authors for revising the manuscript. The methods makes sense now, and I think its quite interesting. While I do have some concerns (e.g. choice of eta, batching may not produce a consistent gradient estimator etc.), I  think the paper should be accepted. I have revised my score accordingly.

That said, the presentation (esp in Section 2) needs to be improved. The main problem is that many symbols have been used without being defined. e.g. phi, q_phi, \pi,  and a few more. While the authors might assume that this is obvious, it can be tricky to a reader - esp. someone like me who is not familiar with GANs. In addition, the derivation of the estimator in Section 3 was also sloppy. There are neater ways to derive this using RKHS theory without doing this on a d' dimensional space.

Revised summary: The authors present a method for estimating the gradient of some training objective for generative models used to sample data, such as GANs. The idea is that this can be used in a training procedure. The idea is based off the Stein's identity, for which the authors propose a kernelized solution. The key insight comes from rewriting the variational lower bound so that we are left with having to compute the gradients w.r.t a random variable and then applying Stein's identity. The authors present applications in Bayesian NNs and GANs.


Summary
================================================================
The authors present a method for estimating the gradient of some training objective
for generative models used to sample data, such as GANs. The idea is that this can be
used in a training procedure. The idea is based off the Stein's identity, for which the
authors propose a kernelized solution. The authors present applications in Bayesian NNs
and GANs.




Detailed Reviews
================================================================

My main concern is what I raised via a comment, for which I have not received a response
as yet. It seems that you want the gradients w.r.t the parameters phi in (3). But the
line immediately after claims that you need the gradients w.r.t the domain of a random
variable z and the subsequent sections focus on the gradients of the log density with
respect to the domain. I am not quite following the connection here.

Also, it doesn't help that many of the symbols on page 2 which elucidates the set up
have not been defined. What are the quantities phi, q, q_phi, epsilon, and pi?

Presentation
- Bottom row in Figure 1 needs to be labeled. I eventually figured that the colors
  correspond to the figures above, but a reader is easily confused.
- As someone who is not familiar with BNNs, I found the description in Section 4.2
  inadequate.

Some practical concerns:
- The fact that we need to construct a kernel matrix is concerning. Have you tried
  batch verstions of these estimator which update the gradients with a few data points?
- How is the parameter \eta chosen in practice? Can you comment on the values that you
  used and how it compared to the eigenvalues of the kernel matrix?

Minor
- What is the purpose behind sections 3.1 and 3.2? They don't seem pertinent to the rest
  of the exposition. Same goes for section 3.5? I don't see the authors using the
  gradient estimators for out-of-sample points?

I am giving an indifferent score mostly because I did not follow most of the details.

---

> ### Author Response · Authors · 2017-12-15
> **thank you for your review and answering your comments**
>
> (We have revised the paper to make the presentation clearer. Please consider it and we would welcome your feedback.)
>
> Thank you for your time for reviewing the paper. Again we are sorry that the presentation is not very clear in the first version of the manuscript. We have revised the paper according to your comments and added a brief introduction to Bayesian neural networks in the appendix.
>
> We believe that our paper is highly novel and contains significant contributions (as reviewer 3 commented). The paper is based on an important observation that an accurate gradient approximation method would be very helpful in many learning tasks that involve fitting an implicit distribution. As the other two reviewers pointed out, the proposed Stein gradient estimator is highly novel, and the experiments consider novel tasks that have not been considered in the literature, e.g. meta-learning for approximate inference, and entropy regularisation methods for GANs.
>
> Now for your detailed comments:
>
> 1. notations of phi, pi, etc.
> We are sorry again for unclear presentation in the first version. In the latest version of the manuscript, we have explicitly defined them and provided a detailed derivation of the entropy gradient in eq (3). Please let us know if it is still unclear.
>
> 2. computing kernel matrix.
> In section 4.3 we performed mini-batch training, and this means we only need to compute the gradient of log q on the mini-batch data. We found that with mini-batch size K=100 (which is typical for deep learning tasks) the computational cost is quite cheap, see the revised paper for a report of running time.
>
> 3. choice of \eta.
> Indeed for kernel methods, \eta needs to be tuned. However, our empirical observation indicates that for better performance of the Stein approach, small \eta is often preferred than large ones. Apparently, matrix inversion has numerical issues, so in our tests, we set \eta to be some small value but large enough to ensure numerical stability.
>
> 4. purpose of 3.1 and 3.2 (in the first version).
> Since the Stein gradient estimator is kernel-based, we need to compare to existing kernel-based gradient estimator. Therefore we introduce them in 3.1 and 3.2 (in the first version of the paper).
>
> 5. purpose of 3.5 (in the first version).
> Our experiment 4.1 actually needs predictive estimators, since we want the particles of parallel chains to be independent of each other. The estimator derived in section 3.3 (of the first version) introduces correlations between the estimates of the score function at different locations.
>
> Also in an on-going work, we apply the proposed Stein gradient estimator to training implicit generative models, which also requires predicting the gradient values. We already have some success on MNIST data, and now we are incorporating kernel learning techniques to scale it to massive data.
>
> Thank you again for reading the feedback, and we look forward to hearing from you again.

---

> ### Author Response · Authors · 2017-12-20
> **thank you for your updated review, and answer your further comments**
>
> Thank you for the positive review. We will think about how to revise the paper. Now on your further comments:
>
> 1. consistency
> We did not claim the proposed Stein gradient estimator is unbiased. This is because: 1) we used the V-statistics of KSD, 2) the fixed point of the MC approximated objective is not necessary the fixed point of the KSD. Similar things apply to the KDE and Score matching estimators. However, asymtotic consistency results have been proved for KDE, and Score matching the proof requires the kernel machine hypothesis set to contain the ground truth. This is not always the case, and our proposal might be prefered here because it is non-parametric.
>
> We are currently working on establishing similar asymtotic consistency results for the Stein gradient estimator.
>
> 2. preference of the RKHS story
> Indeed if we directly start to talk about kernels then I would rather prefer the derivation of section 3.2 (in the current version). However for people (like engineers) who are not familiar with the RKHS theory, the explanation of section 3.1 might be more intuitive, and that's why I decided to include both of them. This is in similar spirit as to derive linear regression equations in many statistics textbooks: we first write down the solutions, and then notice that we can use the kernel trick to address the d' >> K problem.
>
> Thank you for your feedback again and do let us know what we can do to improve the paper.

---

### Official Review · AnonReviewer3 · 2017-11-29
**Interesting paper with novel and significant contribution!**

**Rating:** 7
**Confidence:** 4

**Review:**

In this paper, the authors proposed the Stein gradient estimator, which directly estimates the score function of the implicit distribution. Direct estimation of gradient is crucial in the context of GAN because it could potentially lead to more accurate updates. Motivated by the Stein’s identity, the authors proposed to estimate the gradient term by replacing expectation with the empirical counterpart and then turn the resulting formulation into a regularized regression problem. They also showed that the traditional score matching estimator (Hyvarinen 2005) can be obtained as a special case of their estimator. Moreover, they also showed that their estimator can be obtained by minimizing the kernelized Stein discrepancy (KSD) which has been used in goodness-of-fit test. In the experiments, the proposed method is evaluated on few tasks including Hamiltonian flow with approximate gradients, meta-learning of approximate posterior samplers, and GANs using entropy regularization.

The novelty of this work consists of an approach based on score matching and Stein’s identity to estimate the gradient directly and the empirical results of the proposed method on meta-learning for approximate inference and entropy regularized GANs. The proposed method is new and technically sound. The authors also demonstrated through several experiments that the proposed technique can be applied in a wide range of applications.

Nevertheless, I suspect that the drawback of this method compared to existing ones is computational cost. If it takes significantly longer to compute the gradient using proposed estimator compared to existing methods, the gain in terms of accuracy is questionable. By spending the same amount of time, we may obtain an equally accurate estimate using other methods. For example, the authors claimed in Section 4.3 that the Stein gradient estimator is faster than other methods, but it is not clear as to why this is the case. Hence, the comparison in terms of computational cost should also be included either in the text or in the experiment section.

While the proposed Stein gradient estimator is technically interesting, the experimental results do not seem to evident that it significantly outperforms existing techniques. In Section 4.2, the authors only consider four datasets (out of six UCI datasets). Also, in Section 4.3, it is not clear what the point of this experiment is: whether to show that entropy regularization helps or the Stein gradient estimator outperforms other estimators.

Some comments:

- Perhaps, it is better to move Section 3.3 before Section 3.2 to emphasize the main contribution of this work, i.e., using Stein’s identity to derive an estimate of the gradient of the score function.
- Stein gradient estimator vs KDE: What if the kernel is not translation invariant?
- In Section 4.3, why did you consider the entropy regularizer? How does it help answer the main hypothesis of this paper?
- The experiments in Section 4.3 seems to be a bit out of context.

---

> ### Author Response · Authors · 2017-12-15
> **thank you for the positive review, and answering your comments**
>
> (We have revised the paper to make the presentation clearer. Please consider it and we would welcome your feedback.)
>
> Thank you for your time for reviewing the paper. We appreciate your positive comment that the paper contains significant contributions to the community. The diversity of the experimental tasks show that gradient estimation is fundamental to many machine learning tasks, so we believe the proposed estimator is widely applicable as you pointed out.
>
> Also, we would like to thank you for the suggestions on making the paper clearer. We have re-organised the presentation to emphasise the contribution of the Stein gradient estimator.
>
> Now on your comments:
>
> 1. Computation cost
> We added two paragraphs in the manuscript for further discussions on this. In short, we discussed:
>
> Comparisons between kernel methods and other ideas. It is also known that the denoising auto-encoder (DAE), when trained with infinitesimal noise, also provides a score function estimator. However, this requires training the DAE, and depending on the neural network architecture, it can take significantly much more time compared to the kernel-based estimators which often have analytical solutions.
>
> For the three kernel-based methods mentioned in the paper, both Score and Stein method require inverting a K*K matrix (O(K^3) time). All three methods require computing the kernel matrix (O(K^2 * d) time). However in the BNN and GAN experiments, since d >> K, the cost is dominated by the kernel matrix computation, meaning that all three methods have similar computational costs. Indeed we reported the running times for the GAN experiments which are almost identical. Also adding the entropy regularisation only resulted in 1s/epoch more time compared to vanilla BEGAN, which is actually quite cheap.
>
> 2. BNN experiment.
> We have clearly shown that the Stein approach is significantly better than the other two gradient estimators. SGLD with small step-size is known to work well, and the Stein method works equally well in this case. To our knowledge, this is the first attempt of meta-learning for approximate samplers, and our results demonstrate that this direction is worth investigation. We strongly believe that with a better neural network structure our method can be improved.
>
> Regarding the scale of the experiment: UCI datasets are standard benchmarks for Bayesian neural networks (e.g. see the PBP paper, Hernandez-Lobato and Adams 2015), and for datasets of this scale, we know that point estimates work worse. The size of the network is of the same scale as reported in Fig 5 (left) of (Andrychowicz et al. 2016).
>
> 3. the GAN experiment in 4.3
> The purpose of section 4.3 is to show the application of gradient estimation methods to tasks other than approximate inference (4.1 and 4.2). Our goal here is to show: (i) by adding entropy regulariser it can help address the mode collapse problem, and (ii) the resulting diversity measure also reflects the approximation accuracy of the entropy gradient. In this experiment, we showed that the Stein approach works considerably better.
>
> Indeed our ultimate goal of developing gradient estimation methods is to use them for training implicit generative models, and if successful, it can serve as an alternative to GAN-like approaches. In an on-going work, we already have some success on MNIST data. We are now working on incorporating kernel learning techniques to scale it to massive data.
>
> 4. non translation invariant kernel case.
> To our knowledge, it is rare for KDE methods to use non translation invariant kernels. And we have never seen consistency results proved for KDE gradient estimator in this case. But indeed connections between Stein and KDE methods is still a research question when using non translation invariant kernels.
>
> Thank you again for reading the feedback and we look forward to hearing from you again.

---

### Official Review · AnonReviewer1 · 2017-12-01
**an interesting paper**

**Rating:** 6
**Confidence:** 2

**Review:**

This paper deals with the estimation of the score function, i.e., the derivative of the log likelihood. Some methods were introduced and a new method using Stein identity was proposed. The setup of the trasnductive learning was introduced to add the prediction power to the proposed method. The method was used to several applications.

This is an interesting approach to estimate the score function for location models in a non-parametric way. I have a couple of minor comments below.

- Stein identity is the formula that holds for the class of ellipsoidal distribution including Gaussian distribution. I'm not sure the term "Stein identity" is appropriate to express the equation (8).
- Some boundary condition should be assumed to assure that integration by parts works properly. Describing an explicit boundary condition to guarantee the proper estimation would be nice.

---

> ### Author Response · Authors · 2017-12-15
> **Thank you for your review, and answering your comments**
>
> (We have revised the paper to make the presentation clearer. Please consider it and we would welcome your feedback.)
>
> Thank you for your time for reviewing the paper. We appreciate your comment that the proposed approach is interesting.
>
> We would like to emphasise that our work is highly novel (as both reviewers 2 and 3 pointed out).
>
> 1. The Stein gradient estimator is a novel score function estimator, which, as you mentioned, generalises the score matching estimator. To our knowledge, this is the first **non-parametric** direct estimator: the KDE method, although also non-parametric, is an **indirect** method as it first estimates the density then takes the gradient.
>
> 2. We applied the gradient estimation methods to a wide range of novel applications. To our knowledge, before our development, no paper has considered meta-learning tasks for approximate inference. Also, the entropy regularisation idea for GANs is novel, which cannot be done without an efficient gradient estimation method.
>
> In an on-going work, we have applied the Stein gradient estimator to training implicit generative models, and small-scale experiments have shown promising results.
>
> Now on your comments:
>
> 1. Yes as you pointed out, the original Stein's identity (Stein 1972, 1981) are for Gaussian distributions. However, the identity has been generalised to more general case. In equation (6) of the revised manuscript, we explicitly write down the integration by part derivations with the boundary condition assumed. Indeed for distributions with Gaussian-like tails almost any test function will satisfy the boundary condition.
>
> 2. If you would like to see an counterexample: if q(x) is Cauchy, then h(x) should be less or equal than order of x^2. But in practice, since the kernel in use often has decaying tails, it is generally the case that the boundary condition is satisfied.
>
> Thank you again for reading the feedback and we look forward to hearing from you again.

---

### Comment · AnonReviewer2 · 2017-11-24
**Some Questions**

1. Can you give a reference for Stein's multivariate identity - paper and theorem? The Stein 1981 paper only seems to discuss the univariate case.
2. What are the following quantities on page 2: phi, q, q_phi, epsilon, pi
3. It seems that you want the gradients w.r.t the parameters phi in (3). But the line immediately claims that you need the gradients w.r.t the domain of a random variable z and the subsequent sections focus on the gradients of the log density with respect to the domain. I am not quite following the connection here.

---

> ### Author Response · Authors · 2017-11-30
> **Thanks for your questions!**
>
> Thank you for your time on reviewing this paper!
>
> On your questions:
> 1. Yes the original Stein (1981) paper only described the identity for a multivariate Gaussian distribution, and it assumed the test function to output scalars. However, the proof technique only used integration by parts, which, if the boundary condition is assumed, should be able to generalise to general distribution case.
>
> Our twist of the formula comes from the observation that we can pack multiple (scalar output) test functions into a vector. This has also been considered in e.g. Liu et al. (2016). Will fix the descriptions -- thank you!
>
> 2. Sorry for the rush of background introduction. Here q denotes the approximate posterior, and q_{\phi} just explicitly writes the dependency of q to its parameter \phi. \epsilon is the noise variable used in the reparameterisation trick.
>
> 3. Again sorry for the rush of the derivation -- will add a full equation in appendix.
> In short the idea is to apply the reparameterisation trick, and notice that the gradient of \phi contains the path gradient (the first term in (3)) and an expectation of the REINFORCE gradient (the second term). Using the log-derivative trick we can show that the second term is zero.
>
> Hope this helps!

---

> > ### Comment · AnonReviewer2 · 2017-12-02
> > **Not clear yet**
> >
> > I am still not following the details for 1 and 3 yet. Feel free to elaborate below, but ideally these should have appeared in the paper.
> > For 2, what is pi?

---

> > > ### Author Response · Authors · 2017-12-04
> > > **Quick answer for your questions**
> > >
> > > I am sorry again for rush derivations, will revise the derivations and provide an official response to your review. But just to quickly explain Q1 and Q3 here:
> > >
> > > Q1:
> > > Let's assume h(x) output a scalar for a moment. Then, stein's identity can be proved using integration by parts:
> > >
> > > \int q(x) [ h(x) * dlogq(x)/dx + dh(x)/dx] dx
> > > = \int [h(x) * dq(x)/dx + q(x) * dh(x)/dx ] dx    // dlogx/dx = x^{-1}
> > > = \int d[h(x)q(x)]/dx dx
> > > = h(x)q(x)|_{\partial X}
> > > = 0    // assumed by the boundary condition
> > >
> > > Now write multi-dimension version h(x) = (h_1(x), ..., h_{d'}(x)). Then looking at eq (8), we notice that the ith row of the LHS matrix is actually
> > > \int q(x) [ h_i(x) * dlogq(x)/dx + dh_i(x)/dx] dx,
> > > meaning that if we assume boundary condition for h(x) (which also implies boundary condition for h_i(x)), then we can prove again that the LHS matrix is actually zero.
> > >
> > > Q3:
> > > Here I assume we can apply the reparameterisation trick for q (see the VAE papers). This says,
> > > sampling z ~ q_{\phi}(z | x) is equivalent to 1) sample \epsilon ~ \pi(\epsilon), then 2) compute z = f_{\phi}(\epsilon, x). \pi(\epsilon) is the distribution of the noise which is usually Gaussian. This means, we can rewrite the expectation in q to expectation in \pi. Please see section 2.4 in the original VAE paper (Kingma and Welling 2013) for an example math derivation.
> > >
> > > Then I wanted to differentiate the variational lower-bound wrt \phi. Especially, eq. (3) derived the gradient of the entropy term wrt \phi. I will add in detailed derivations in revision, but for your quick reference please see eq (5-7) in (Roeder et al. 2017) for an example derivation.

---

### Public Comment · (anonymous) · 2017-11-30
**an interesting paper**

This paper deals with the estimation of the score function, i.e., the derivative of the log likelihood. Some methods were introduced and a new method using Stein identity was proposed. The setup of the trasnductive learning was introduced to add the prediction power to the proposed method. The method was used to several applications.

This is an interesting approach to estimate the score function for location models in a non-parametric way. I have a couple of minor comments below.

- Stein identity is the formula that holds for the class of ellipsoidal distribution including Gaussian distribution. I'm not sure the term "Stein identity" is appropriate to express the equation (8).
- Some boundary condition should be assumed to assure that integration by parts works properly. Describing an explicit boundary condition to guarantee the proper estimation would be nice.

---

> ### Author Response · Authors · 2017-11-30
> **Thanks for your comments!**
>
> Thank you for your interest in this paper!
>
> For your questions:
> 1. Yes the original Stein (1981) paper only described the identity for a multivariate Gaussian distribution, and it assumed the test function to output scalars. However, the proof technique only used integration by parts, which, if the boundary condition is assumed, should be able to generalise to general distribution case.
>
> Our twist of the formula comes from the observation that we can pack multiple (scalar output) test functions into a vector. This has also been considered in e.g. Liu et al. (2016).
>
> What would you suggest to call (8) other than "Stein's identity"?
>
> 2. Yes it would be nice to have an example, however in the Gaussian case since the tails decay exponentially, almost all functions satisfy the boundary condition. Maybe it would be helpful to do an example with long-tail distributions.
>
> Hope this helps and thanks again!

---

### Author Response · Authors · 2018-02-23
**Updated results in section 4.2**

We found a bug related to data sub-sampling in the original experiments, so we fixed the bug and conducted the experiments again. Now score matching methods works much better than before, but it is still the worst method. The proposed Stein gradient estimator approach remains the best.

=================================
Typo: In Figure 3, the sonar panels should have x axis ranging from 0 to 5000. We have corrected this in the arXiv version https://arxiv.org/abs/1705.07107

---

### Decision · Program_Chairs · 2018-01-29
**ICLR 2018 Conference Acceptance Decision**

**Decision:**

Accept (Poster)

**Comment:**

The paper presents the Stein gradient estimator, a kernelized direct estimate of the score function for implicitly defined models. The authors demonstrate the estimator for GANs, meta-learning for approx. inference in Bayesian NNs, and approximating gradient-free MCMC. The reviewers found the method interesting and principled.  The GAN experiments are somewhat toy-ish as far as I am concerned, so I'd encourage the authors to try out larger-scale models if possible, but otherwise this should be an interesting addition to ICLR.